# *Clostridia* and Enteroviruses as Synergistic Triggers of Type 1 Diabetes Mellitus

**DOI:** 10.3390/ijms24098336

**Published:** 2023-05-06

**Authors:** Robert Root-Bernstein, Kaylie Chiles, Jack Huber, Alison Ziehl, Miah Turke, Maja Pietrowicz

**Affiliations:** 1Department of Physiology, Michigan State University, East Lansing, MI 48824, USA; chileska@msu.edu (K.C.); huberja2@msu.edu (J.H.); ziehlali@msu.edu (A.Z.); pietro10@msu.edu (M.P.); 2Department of Microbiology and Molecular Genetics, Michigan State University, East Lansing, MI 48824, USA; 3Department of Chemistry, University of Chicago, Chicago, IL 60637, USA; mturke@uchicago.edu

**Keywords:** diabetes, COX, *Clostridium*, complementary antigens, idiotype–anti-idiotype, circulating immune complexes, T cell receptors, synergism

## Abstract

What triggers type 1 diabetes mellitus (T1DM)? One common assumption is that triggers are individual microbes that mimic autoantibody targets such as insulin (INS). However, most microbes highly associated with T1DM pathogenesis, such as coxsackieviruses (COX), lack INS mimicry and have failed to induce T1DM in animal models. Using proteomic similarity search techniques, we found that COX actually mimicked the INS receptor (INSR). *Clostridia* were the best mimics of INS. *Clostridia* antibodies cross-reacted with INS in ELISA experiments, confirming mimicry. COX antibodies cross-reacted with INSR. *Clostridia* antibodies further bound to COX antibodies as idiotype–anti-idiotype pairs conserving INS–INSR complementarity. Ultraviolet spectrometry studies demonstrated that INS-like *Clostridia* peptides bound to INSR-like COX peptides. These complementary peptides were also recognized as antigens by T cell receptor sequences derived from T1DM patients. Finally, most sera from T1DM patients bound strongly to inactivated *Clostridium sporogenes*, while most sera from healthy individuals did not; T1DM sera also exhibited evidence of anti-idiotype antibodies against idiotypic INS, glutamic acid decarboxylase, and protein tyrosine phosphatase non-receptor (islet antigen-2) antibodies. These results suggest that T1DM is triggered by combined enterovirus-*Clostridium* (and possibly combined Epstein–Barr-virus-*Streptococcal*) infections, and the probable rate of such co-infections approximates the rate of new T1DM diagnoses.

## 1. Introduction

Type 1 diabetes mellitus (T1DM) is an autoimmune disease in which antibodies and T cells target a range of host autoantigens associated with beta cell production of insulin (INS), resulting in loss of INS production and consequent hyperglycemia. While much of the focus of T1DM research is on the autoimmune targeting of INS itself [1,2,3], antibodies and T cells also target glutamic acid decarboxylases (GAD) [1,2,3], protein tyrosine phosphatase non-receptor types (related to islet-associated protein or PTPN-IA-2) [1,2,3], the INS receptor (INSR) [4,5,6,7,8], and glucagon [9,10]. This combination of autoantigenic targets helps to explain why pancreatic beta cells are particular targets of T1DM pathogenesis. However, the major mystery concerning T1DM pathogenesis is the disease’s etiology: what triggers the autoimmunity directed at these pancreatic targets?

Determining the causes of autoimmune diseases such as T1DM has turned out to be a recalcitrant problem. Despite over a century of epidemiological and experimental studies of autoimmunity, the natural cause of no human autoimmune disease has yet to be discovered. It is generally believed that predisposition to autoimmune diseases is determined by genetic factors but that infectious (or other environmental) factors are required to trigger the disease process (e.g., [11,12,13,14,15,16,17]). Epidemiological methods in conjunction with individual patient case reports are generally used to try to identify what these infectious triggers may be. The general assumption is that causative microbes present antigens to the immune system that mimic the host autoantigens that the disease subsequently targets.

The onset of T1DM has been associated epidemiologically with both viral and bacterial infections, and the best clinical correlations for the onset of T1DM are probably the coxsackieviruses (COX), both A and B strains [18,19,20,21,22,23]. However, other enteroviruses [18,23,24,25], such as rubella, mumps, rotaviruses, cytomegalovirus (CMV), Epstein–Barr virus (EBV), and hepatitis C virus (HCV), have also been associated with T1DM initiation [13,26,27,28,29]. Bacterial infections associated with onset of T1DM include *Bordatella pertussis* and *Mycobacterium* species [30,31] and *Helicobacter pylori* [32]. Studies using T cells specific for the INS B chain, which is often considered to be the main target of T1DM autoimmunity, have identified *Streptococci*, *Clostridia*, *Escherichia coli*, and *Pseudomonas* [33] as potential mimics. T cells specific for GAD65 identified *Streptococci*, *Staphylococci*, *Haemophilus*, *Legionella*, and *Chlamydia* as the most likely triggers [34]. Most recently, significant differences in gut microbiota between children who have just been diagnosed with T1DM and those who have not suggest that intestinal bacteria may also play a critical role in triggering or regulating the development of diabetes [35,36,37,38,39,40]. The focus on the gut microbiome has led to the identification of *Parabacteroides distasonis* as a possible trigger of T1DM because its bacterial antigens activated both human T cell clones from T1DM patients and T cell hybridomas from nonobese diabetic (NOD) mice specific to the INS B chain residues 9–23 [40]. However, *P. distasonis* was not identified by previous studies of T cells reactive to INS [41,42]. Another study identified peptides from *Bacteroides fragilis* and *Clostridium asparigiforme* as potent activators of human T1DM T cells responsive to pre-pro-INS [41], of which only *Clostridia* were identified in previous studies [41,42].

Unfortunately, the numerous agents associated with T1DM leave significant questions regarding the sufficiency and necessity of any one microbe as a trigger for diabetes, a problem that has persisted for decades [28,43]. Attempts to model the onset of T1DM using individual infectious agents from the list above have thus far failed. No one has been able to produce T1DM in any animal using any of the single infectious agent listed above. COX and CMV exacerbate or accelerate disease in rodents already producing autoantibodies (e.g., NOD mice) or pretreated with the pancreatic toxin streptozotocin [21,44] but produce only transient pancreatic pathology in select strains of non-diabetic mice [45]. Monkeys infected with coxsackie B virus types 3 and 4 also develop transient pancreatitis but fail to develop chronic diabetes [46]. Cross-reactivity between COX antigens (strains B1-B6) and human T1DM antibodies reactive to either pro-INS or GAD could not be identified [47,48,49,50,51,52,53]. COX antibodies do not appear to be cross-reactive with INS, nor are INS antibodies cross-reactive with COX [54]. Moreover, although evidence of COX infections appears in temporal relationships with subsequent T1DM diagnosis [50,55,56], at least one live enterovirus vaccine—oral polio—is not associated with any increased risk of T1DM, even among genetically high-risk individuals [57,58]. However, COX infections have been linked to the development of INS receptor antibodies [59].

Even more confusingly, some putative triggers of T1DM have actually prevented the development of T1DM in animal models (reviewed in [60]). For example, pertussis vaccine protected streptozotocin-treated CD-1 mice against developing diabetes [61]; immunization with *Mycobacterium. leprae* [62], Bacillus Calmette-Guerin (BCG) [63], or *Clostridium butyricum* [64] also prevented diabetes in NOD mice [65].

Taken as a whole, the results summarized above demonstrate that mono-infectious approaches to modeling T1DM have universally failed or yielded results that seem to contradict the role of any particular microbe as a cause of the disease. These failures led Horwitz et al. [20,66,67] to suggest that the role of COX may not be as direct triggers of T1DM via molecular mimicry but rather as bystander infections supporting some other infectious agent. As Filippi and von Herrath [68] suggested, “This could be explained by the fact that viral association with T1D will likely be multifactorial”.

So, perhaps the difficulty identifying “the cause” of T1DM stems from the assumption that “the cause” is mono-factorial and resides in the unique identification of one of the microbes listed above as “the” T1DM trigger. However, perhaps no single microbe is both necessary and sufficient. Perhaps a new paradigm is required, based on multiple, concurrent infections as autoimmune disease initiators stimulating the production of complementary or synergistic sets of autoantibodies and TCR directed at multiple targets simultaneously.

The major aim of our research is to explore the possibility that sets of microbes may cooperate to induce T1DM. This aim requires a shift in the types of experiments and logic employed to test possible T1DM etiologies. The standard approach to elucidating autoimmune disease etiology is based on Koch’s postulates, which assume a single etiological agent. A concurrent-infection model requires that two or more etiological agents be involved. The use of multiple agents is consistent with the observation that there are multiple autoantigen targets in T1DM. Therefore, one aim of our research is to use proteonomic methods to identify microbes that mimic the key autoantigens in T1DM, INS, and INSR. A second aim is to use the mimicry data as the basis for experimentally testing whether antibodies against the identified microbes cross-react with INS and INSR. Because INS and INSR are molecularly complementary, logically it follows that the microbial antigens inducing cross-reactive immunity to INS and INSR will also be complementary to each other. Thus, a third aim is to test for possible antigen complementarity. Three such tests are presented. The first test is based on the proposition that some microbial antigens mimicking INS and INSR will bind to each other just as INS and INSR themselves bind to each other. A second test is predicated on the assumption that antibodies against these INS and INSR mimics will act like idiotype–anti-idiotype pairs. A third test is whether or not T cell receptor sequences from T1DM patients that have previously been demonstrated to bind to INS and INSR peptides also recognize these microbial antigens. The final aim is to test whether sera from T1DM patients recognize the microbial antigens identified in testing the previous aims.

The results consistently reveal that COX mimic INSR antigens; *Clostridia* mimic INS; and that the resulting immune responses whether based on animal-derived antibodies, human TCR sequences, or T1DM sera, involve idiotype–anti-idiotype relationships to sets of complementary antigens. No set of control antibodies displayed a similar range of interactions. These results suggest a new role for COX and other enteroviruses (inducing INSR, rather than INS, cross-reactivity) in T1DM etiology; identify Clostridia as the triggers of INS cross-reactivity for the first time; and provide the first evidence suggesting a multifactorial, synergistic mechanism for T1DM etiology.

## 2. Results

### 2.1. Proteomic Search for Microbial Similarities to T1DM Autoantigens

Our investigation began with a simple strategy, which was to search for the best similarity matches (as defined by high rates of identity over short regions of ten or twelve amino acids identified using BLOSUM80) between INS, INSR, and the bacteria and viruses listed in the UniProtKB databases of microbial proteins. We relied on the internal statistical algorithms of BLASTP to provide the best matches. Matches involving microbes that do not infect or that live as commensal organisms in human beings were eliminated from the search results as being unlikely to induce active immunity. Figure 1 and Figure 2 display the key findings.

As Figure 1 and Figure 2 demonstrate, only *Clostridium* species repeatedly appeared as significantly similar to human INS A and B chains. A more targeted similarity search with BLASTP comparing INS A and B chains with pathogenic *Clostridia* (*C. clostridioforme, C. difficile, C. perfingens, C. sordelli, C. tetani*) revealed a much broader set of INS similarities in each case. Only the *C. difficile* similarities are presented here as a representative case (Appendix B). Similarities to the INS B chain were much more common than those to the INS A chain, which is consistent with the B chain being the main target of autoantibodies in T1DM. No significant similarities (defined as at least six identities in a sequence of ten amino acids) were found between any virus, including COX, and either the INS A or B chains.

Because COX has very consistently been associated with T1DM using many types of methods [18,19,20,21,22,23,24,25], the possibility that it was responsible for triggering INSR rather than INS cross-reactivity was investigated. A BLASTP search of INSR against the entire SwissProt virus database revealed that among the top 100 virus matches, only five types of human viruses had significant similarities to INSR: Human herpesviruses (HHV) 1, 2, 3, and 6 and coxsackieviruses (Appendix A). HHV1, 2, and 6 are not among the viruses frequently associated epidemiologically with T1DM (see Section 1), while HHV3 (varicella zoster virus) is almost certainly not a significant trigger of T1DM, as demonstrated by the fact that T1DM cases continued to increase following almost universal vaccination against this virus. Thus, we focused our further efforts on coxsackieviruses for which many very significant INSR similarities were found. The results for type B4 are typical of B-type COX and are provided in Figure 3, while the lesser number of significant matches for COX A16 (typical of A-type COX) are illustrated in Appendix B.

BLASTP similarity searches were also performed on protein tyrosine phosphatase non-receptor type 1 (PTPN(IA-2)) and glutamic acid decarboxylase 65 (GAD65) because these are the two most common intracellular T1DM targets (Appendix C). Many significant similarities exist between *C. difficile* proteins and both PTPN(IA-2) and GAD65. No significant similarities were found between COX and PTPN(IA-2), and only three similarities were found to GAD65 (Appendix C). Thus, *Clostridia* are much more likely triggers of anti-PTPRN and anti-GAD65 antibodies or TCR than are COX.

In sum, pathogenic *Clostridia* were found to have the most abundant similarities to human INS of any bacteria associated with human health that are currently in the SwissProtKB database. The only other bacteria to rise to an equivalent degree of similarity were *Lactobacilli*, *Bacteroides fragilis*, *Mycobacterium goodie*, and *Pseudomonas* species, but these are all commensal components of the human microbiome and unlikely to trigger autoimmunity. *Clostridia* also displayed many significant similarities to two other major targets of autoimmunity in T1DM, PTPN(IA2), and GAD65. COX did not display significant similarities to INS or PTPN(IA2), and only three to GAD65, but it did display many significant similarities to INSR. These results suggest that COX may initiate an immune response to INSR rather than to INS, while *Clostridia*, which have not previously been investigated in this regard, may trigger the observed responses to INS, PTPN(IA2), and/or GAD65. According to the results, some herpes viruses may also contribute to T1DM risk. These predictions are tested in the next sections.

### 2.2. Microbial Antibody Binding to INS, INSR, and Peptides Derived from INSR

As a first test of the proteomic search results, we explored the ability of antibodies against COX, *Clostridia*, and a range of other viruses and bacteria to bind to INS, INSR, and peptides derived from the INSR using quantitative enzyme-linked immunoadsorption assays (ELISA). The regions tested for antibody binding were in the extracellular portion of the receptor; although some of the COX-INSR similarities reported in the previous section were in the transmembrane and intracellular portions of the receptor, these were assumed to be inaccessible to antibodies. The results generally confirmed the proteomic findings and are summarized in Figure 4 and Figure 5. COX antibodies, whether derived from monkey, horse, or mouse, consistently recognized INSR as a target with significant affinity, while *Clostridium* antibodies did not (Figure 4). In general, COX antibodies were specific for regions of INSR that are associated with INS binding. This observation may have two different kinds of importance. One is that COX binding to INSR may interfere with INS binding during T1DM, producing some degree of INS resistance. Secondly, the specificity for INS-binding regions may indicate that some COX antibodies mimic INS itself. This possibility is tested below. Note, however, that COX antibodies derived from different species displayed varying patterns of INSR peptide specificities, suggesting that INSR autoantigenicity may have a genetic component. Note also that although some *Clostridium* antibodies recognized one INSR region (amino acid positions 897–915)—apparently a highly cross-reactive sequence (see Figure 4)—this cross-reactivity was not sufficient to result in recognition by these *Clostridium* antibodies (nor Epstein–Barr virus antibodies—Figure 5) also cross-reacting with whole INSR, suggesting that this particular region is perhaps not very accessible to antibodies. None of the antibodies tested recognized glucagon, glucagon receptor peptides, or the beta 2 adrenergic receptor (Figure 4).

Figure 5 illustrates the fact that no other virus antibody tested (influenza A and B viruses, cytomegalovirus, adenoviruses, or herpes simplex virus type 1), except for the Epstein–Barr virus (EBV), cross-reacted with INS, glucagon, or INSR. EBV antibodies recognized INSR 897–915 and the whole INSR. Of the bacterial antibodies tested (*Enterococci, Pseudomonas aeruginosa*, *Mycobacterium tuberculosis*, *Staphylococcus aureus*, and group A *Streptococci*), only group A *Streptococci* recognized INS as a target, and once again, while *P. aeruginosa* and *M. tuberculosis* recognized several INSR peptides, they did not bind significantly to whole INSR. None of the antibodies tested recognized glucagon, glucagon receptor peptides, or the beta 2 adrenergic receptor.

Figure 6 and Appendix D provide examples of many of the key results summarized in Figure 4 and Figure 5 (Kd values) as well as a wide range of negative results. These results clearly demonstrate that cross-reactivity is not a function of the species source of the antibodies. COX, regardless of the species in which the antibodies were raised, and enterovirus antibodies uniformly bound INSR and some of its peptide regions. Figure 7 expands on the data summarized in Figure 4 by demonstrating that *Clostridia* antibodies cross-react with INS A and B chains (Kd > 15 µM), though with less affinity than to intact INS. *Clostridia* antibodies bind to INS with both high affinity (Kd 10 nM ) and lower affinity (1.5 µM) interaction.

### 2.3. Complementarity between Virus Antibodies and Bacterial Antibodies

Because *Clostridia* antibodies cross-react with INS and COX/enterovirus antibodies cross-react with INSR, and because INS and INSR are molecularly complementary to each other, one implication is that some *Clostridia* antibodies will be complementary to some COX antibodies, i.e., they will interact like idiotype–anti-idiotype pairs. This possibility was investigated using double-antibody ELISA (DA-ELISA) in which one antibody replaces the antigen in a standard ELISA and is bound to the ELISA plate following serial dilution; a second antibody induced in a different species of animal than the first is then applied at a constant concentration, and a horseradish peroxidase-labeled antibody against the species of the second antibody is then used to determine whether the second antibody has bound to the first. In some cases, a horseradish peroxidase-labelled (HRP) antibody was used for the second antibody, making it possible to measure binding to an antibody derived from the same species as the HRP-labelled antibody. We predicted that some *Clostridia* antibodies would bind specifically to some COX antibodies. The results are summarized in Figure 8 and Figure 9 and illustrated in Figure 10 and Appendix E.

**Figure 8 ijms-24-08336-f008:**
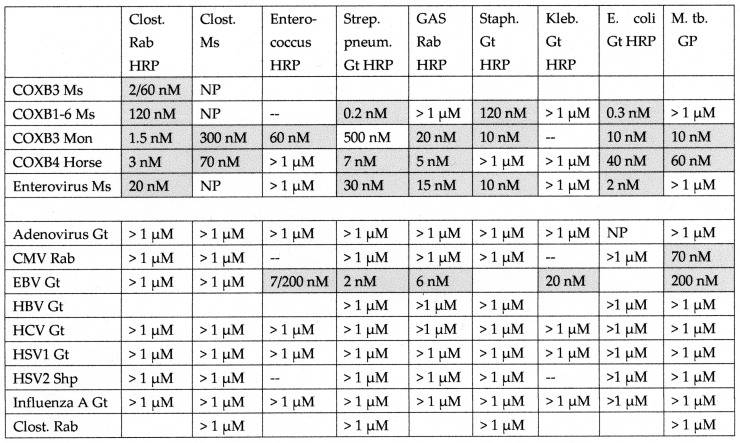
Summary of the binding constants (Kd) from double-antibody ELISA experiments involving a virus antibody potentially binding to a bacterial antibody. See Figure 10 and Appendix E for examples of the binding curves. The binding constants were derived from the inflection points of the curves. COXB = COX type B; CMV = cytomegalovirus; HBV = hepatitis B virus; HCV = hepatitis C virus; HSV = herpes simplex virus; Clost = *Clostridium*; Strep. pneum. = *Streptococcus pneumoniae*; GAS = group A *Streptococci*; Staph = *Staphylococcus aureus*; Kleb. = *Klebsiella*; *E. coli* = *Escherichia coli*; M. tb. = Mycobacterium tuberculosis; Rab = rabbit; Ms = mouse; Gt = goat; Shp = sheep. Where two numbers appear in any particular box, this indicates that the binding curve was doubly inflected, the first number providing the binding constant of the high affinity part of the curve, and the second number providing the binding constant of the low affinity part of the curve. NP = combination not possible.

Briefly, all of the COX antibodies tested bound significantly (i.e., with nanomolar affinity) to at least one *Clostridium* antibody (not all permutations could be tested because some of the antibodies against each microbe were derived from mice). None of the COX antibodies bound to *Klebsiella* antibodies and only one bound to *Enterococcus* antibodies, but most bound to *Streptococcal*, *Staphylococcal*, *Escherichia coli (E. coli)*, and *Mycobacterium tuberculosis (Mtb)* antibodies. *Clostridia* antibodies were more selective, binding only to COX and enterovirus antibodies but not to any of the antibodies against adenoviruses, cytomegalovirus (CMV), Epstein–Barr virus (EBV), hepatitis C virus, herpes simplex type 1, herpes simplex type 2, influenza A virus, or the SARS-CoV-2 spike protein. Of these control antibodies, only those against EBV and SARS-CoV-2 bound significantly to several of the bacterial antibodies (*Enterococci*, *Streptococci*, *Staphylococci*, *Klebsiella*, and *E. coli*). The many negative interactions observed for most of the antibody combinations summarized in Figure 8 and Figure 9 illustrate the point that binding is not associated with the species of animal in which the antibodies are raised and, additionally, that virus–bacteria antibody interactions appear to be much more common than virus–virus antibody interactions (Figure 9). No concerted effort was made to investigate bacteria–bacteria antibody interactions, but some results are provided in Figure 8.

Examples of the binding curves from which Figure 8 and Figure 9 were derived are provided in Figure 10 and Appendix E. These emphasize the specificity of the binding between COX and Clostridia antibodies, although it is also notable that Epstein–Barr virus (EBV) antibodies bound significantly to a number of bacterial antibodies (Figure 9 and Figure 10), particularly in light of the fact that EBV antibodies also recognized INSR and two of the INSR peptides (Figure 10). Notably, one of the bacterial antibodies to which EBV antibodies bound—*Streptococcal* antibodies—also cross-reacted with INS (Figure 10). Thus, a combination of EBV–*Streptococci* may mimic the COX–*Clostridia* combination in terms of recognizing both INS and INSR, thereby resulting in antibodies that have an idiotype–anti-idiotype relationship.

### 2.4. Idiotype–Anti-Idiotype Relationships between Microbial Antibodies and INS, INS Receptor, GAD-65, or PTPN(IA-2) Antibodies

One limitation of the previous set of DA-ELISA experiments is that they do not demonstrate directly that binding between virus and bacterial antibodies involves recognition of INSR and INS epitopes. One way to determine whether the complementarity specifically may involve INSR and INS epitopes is to utilize specific INSR and INS antibodies in place of either a virus or bacterial antibody. In the following experiments, COX and *Clostridia* antibodies were therefore tested for their ability to recognize INSR or INS antibodies as well as antibodies to GAD-65 and PTPN(IA-2).

All of the COX antibodies recognized the INS antibody, but none of the *Clostridia* antibodies did so (Figure 11). Conversely, most of the COX antibodies failed to recognize INSR antibodies (Figure 11), while *Clostridia* antibodies bound to all of the INSR antibodies (Figure 11). These results are consistent with COX mimicking INSR antigens, so that its antibodies mimic INS and therefore bind to antibodies against INS (which mimic INSR). These results are also consistent with *Clostridia* mimicking INS, so that its antibodies mimic INSR and bind to antibodies against INSR (which mimic INS). The weak binding of monkey anti-COX antibodies to INSR antibodies may be due to the fact that several of the regions of the INSR that its antibodies recognize are INS mimics [69,70].

Additionally, COX antibodies bound to GAD-65 and PTPN(IA-2) antibodies, but *Clostridia* antibodies did not (Figure 11). This result is once again consistent with the proteonomic data provided above, indicating that *Clostridia* proteins mimic GAD-65 and PTPN(IA-2) but COX proteins do not (Appendix C). Antibodies against GAD-65 and PTPN(IA-2) should therefore behave like antibodies against *Clostridia*. Thus, *Clostridia* antibodies mimic the behavior of INS antibodies, which in turn mimic the behavior of GAD-65 antibodies and PTPN(IA-2) antibodies, in terms of binding to COX antibodies.

### 2.5. Complementarity of COX and Clostridium Antigens

Because some COX antigens mimic INSR sequences and some *Clostridia* antigens mimic INS, we synthesized several microbial peptide sequences that mimicked these human proteins and used ultraviolet spectroscopy to determine whether they bound to each other, as would be the case if they were complementary antigens. Figure 12 provides the peptide sequences and their similarities to either INS or INSR. *Clostridium* similarities to the INS A chain were specifically chosen because the role of the A chain in the induction of T1DM is generally ignored in favor of the B chain. Figure 12 also demonstrates that a COX peptide mimicking INSR does bind to both *Clostridium* (INS A chain mimic) peptides.

The object of this particular set of experiments was not to provide a complete investigation of the range of potential complementarity peptides or proteins that may exist in COX and *Clostridia* antigens, but merely to test one of the more unusual predictions that follows from the previous results. The fact that this small sample of peptides did yield evidence of complementary antigens suggests the need for a much broader and more complete analysis of COX–*Clostridium* antigen complementarities.

### 2.6. T1DM T Cell Receptor Recognition of COX and Clostridium Antigens

Previous investigators have sequenced the T cell receptor (TCR) sequences that are expanded in T1DM patients with particular specificities for INS or INSR [61,62]. In light of the results above demonstrating that INS-like peptides from *Clostridia* and INSR-like peptides from COX can mimic INS–INSR binding, we investigated whether TCR recognizing INS or INSR would also recognize these microbially-derived peptides. The TCR sequences were synthesized (sequences provided in Table 1). TCR 1, 2, and 4 were from one patient [61]; TCR 8, 9, and 10 were from a second patient [71]; and TCR K2.4, K2.12, and K2.16 were from a third [72]. Ultraviolet light spectrometry was used to determine whether each TCR sequence recognized the COX and *Clostridia* peptides used in the previous set of experiments (Figure 12). The results are shown in Table 1, Figure 13, and Appendix F.

It has previously been demonstrated that these TCR occur in complementary groups for the individual patients from whom they are derived, and that at least two of these TCR pairs bind to each other within each patient group, therefore acting like idiotype–anti-idiotype TCR [73,74]. Thus, the fact that the COX peptide, which mimics INSR, binds best to TCR that mimic INS (TCR 9 and TCR 4, 8,and 9) (Table 1) confirms the mimicry of the peptide for INSR. Notably, other INSR peptides (105–118 and 897–915) also bind to these TCR in a very similar pattern. In contrast, the *Clostridium* peptides mimic INS binding to the INSR-like TCR sequences. The data on INS and INSR peptide binding to these TCR is derived from our previous study [71]. In sum, the hypervariable regions of TCR derived from clones expanded during T1DM recognize a COX peptide similarly to their recognition of INSR peptides, while these TCR recognize *Clostridium* peptides similarly to the way they do INS. These results are consistent with the other experiments described above.

### 2.7. Human T1DM Sera Binding to INS and Clostridium sporogenes

The final set of experiments we performed utilized sera from human T1DM patients, type 2 diabetic patients, and healthy human controls. These sera were tested first for their ability to bind to INS. All of the T1DM sera did so, but healthy individuals and type 2 diabetic sera displayed significantly less INS binding (at least an order of magnitude, and often more than two orders of magnitude less than the T1DM sera) (Appendix G).

The sera were then tested for binding to inactivated *Clostridium sporogenes* that is being developed by GN Neutriceuticals SDN BHD in Malaysia in collaboration with Nanyang Technological University Singapore as a possible cancer therapy [75]. These *C. sporogenes* experiments were run in two different ways that gave equivalent results. The first (Figure 14, LEFT) was to vary the concentration of the *Clostridium* antigen from 10 mg/mL to less than a thousandth of a mg/mL, keeping the serum concentration constant at 1/100. All T1DM sera bound to the *Clostridium* antigen in a generally linear concentration-dependent manner that displayed binding to antigen concentrations significantly less than 1/10,000 mg/mL while healthy control and type 2 diabetic sera displayed binding only at the highest concentrations of antigen (above 1/10 mg/mL).

**Figure 14 ijms-24-08336-f014:**
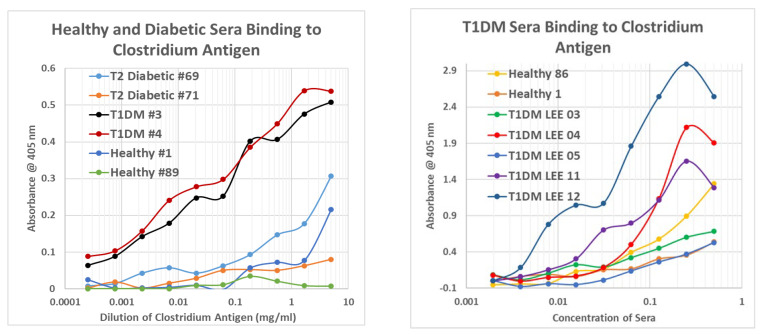
Results of ELISA experiments demonstrating binding of sera from type 1 diabetic patients (T1DM), type 2 diabetic patients (T2 diabetic), and healthy controls to inactivated *Clostridium sporogenes* antigen. LEFT: The sera were held constant and the concentration of *Closridium* antigen was varied. RIGHT: the concentration of *Clostridium* antigen was held constant and concentration of sera was varied.

The second method involved using a constant concentration of *Clostridium* antigen (0.1 mg/mL) and varying the concentration of the sera by serial dilution (Figure 14, RIGHT). These experiments yielded classic S-shaped binding curves. In these experiments, while most of the T1DM sera bound significantly more to the antigen with curves that had lower binding constants, a few of the sera (Figure 14, RIGHT) that had been treated with ethylenediaminetetraacetic acid (EDTA) to prevent clotting did not bind the antigen better than did some of the healthy controls, at least one of which demonstrated significant binding to the antigen. Additional data are presented in Appendix G, and results are summarized in Figure 15.

**Figure 15 ijms-24-08336-f015:**
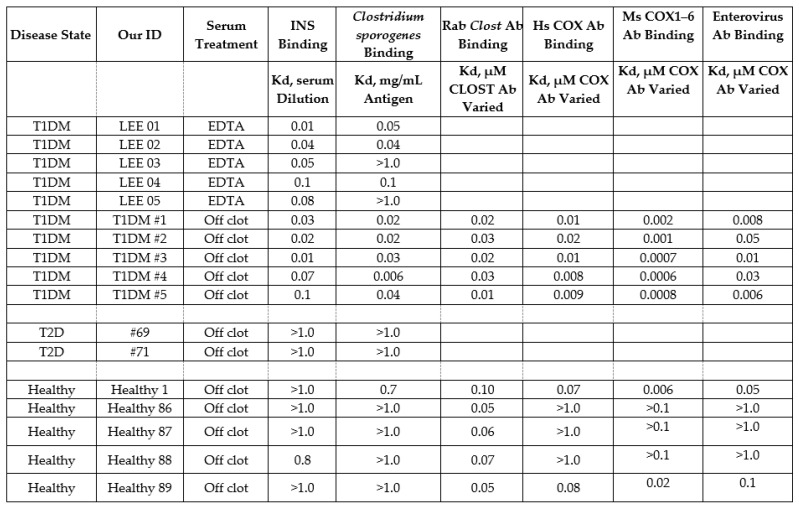
Summary of human sera binding to insulin (INS), *Clostridium sporogenes* antigen, rabbit (Rab) anti-*Clostridium* antibody (Ab), horse (Hs) and mouse (Ms) anti-coxsackievirus (COX) antibodies, and enterovirus antibodies. EDTA = etheylenediaminetetraacetic acid.

There are three possibilities to consider in interpreting these data. One is that the EDTA processing of the one set of T1DM sera may have interfered with antigen recognition. This possibility is given some credence by the fact that most of the T1DM off-the-clot sera displayed lower binding constants (indicating better affinity) than the EDTA-treated sera. Another possibility is that *Clostridia* were not involved in triggering some of the T1DM cases so that their sera lacked *Clostridia* antibodies. A third possibility is that nearly everyone is exposed to *Clostridia* at some time in their lives so that most develop lasting antibodies to the species that wane in some T1DM patients and are robust in some type 2 and healthy individuals. These possibilities are not mutually exclusive.

These data, combined with the TCR data above, are the first evidence clearly linking robust immune responses to *Clostridia* antigens in most T1DM patients and few non-T1DM patients. The data do not exclude the possibility that other microbes are also involved in triggering T1DM, and they are compatible with the likelihood that some people exposed to *Clostridium* antigens do not go on to develop T1DM. There is more detail on these points in Section 3 below.

### 2.8. Human Sera Binding to COX and Clostridia Antibodies

Because we were unable to locate a source of reasonably pure COX antigen, we devised a pair of work-around experiments. On the basis that some *Clostridium* antigens are complementary to COX antigens (Table 1 and Figure 13), that some *Clostridium* antibodies are complementary to COX antibodies (Figure 8 and Figure 10), and that T1DM sera contain antibodies against *Clostridia* (Figure 14 and Figure 15), it follows that T1DM sera should contain antibodies that react to COX antibodies (i.e., are anti-idiotypic to COX antibodies). In fact, T1DM sera do contain antibodies that bind to COX antibodies (Figure 15 and Figure 16). These sera also contain antibodies that recognize *Clostridium* antibodies (Figure 15 and Appendix G), suggesting that these T1DM sera contain anti-idiotype antibodies derived from idiotypic responses to both COX and *Clostridia* antigens. Binding to these COX and *Clostridium* antibodies was much less for healthy individuals than for T1DM patients, with a few exceptions, as illustrated in Figure 15 and Figure 16 and Appendix G.

**Figure 16 ijms-24-08336-f016:**
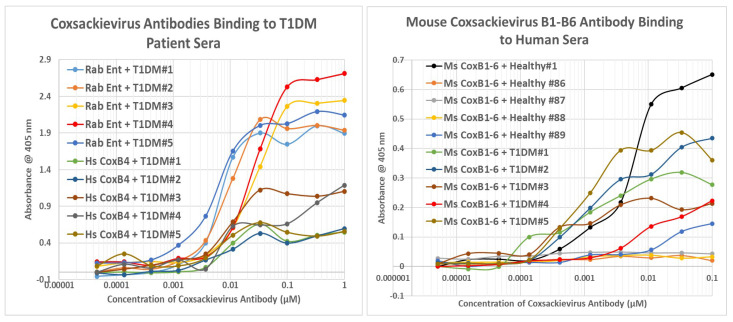
Results of double antibody ELISA experiments involving the binding of human healthy or type 1 diabetic patient (T1DM) sera to enterovirus (Ent) or COX (Cox) antibodies derived from rabbit (Rab), horse (Hs), or mouse (Ms).

## 3. Discussion

### 3.1. Detailed Summary and Interpretation of Results

Section 2.1 demonstrated that pathogenic *Clostridia* species mimic INS, PTPN(IA2), and GAD65 better than any other human pathogen or commensal microbe, while COX was the most likely T1DM-associated mimic of INSR, a result consistent with a previous similarity study that found very significant similarities between INS and PTPN(IA-2) [74]. *Lactobacilli* and *Bifidobacteria* might play a similar role, but their lack of pathogenicity may mediate their autoimmunogenic potential. Section 2.2 demonstrated that, as predicted, COX antibodies bound to INSR and some of its constituent peptides, while *Clostridia* antibodies (and most of the other microbial antibodies tested) did not. Conversely, *Clostridia* antibodies bound to INS, but COX antibodies (and most of the other microbial antibodies tested) did not. Very few other microbial antibodies tested had any affinity for either INS or INSR. The notable exceptions were *Streptococcal* antibodies binding to INS and Epstein–Barr virus (EBV) antibodies recognizing INSR peptides. Notably, COX antibodies were demonstrated to be complementary to *Clostridium* antibodies and EBV antibodies to *Staphylococcal* antibodies, mimicking INS–INSR complementarity (Section 2.3).

Section 2.4 then demonstrated that because COX antibodies mimic INS by binding to INSR, COX antibodies also bind to INS antibodies. Similarly, because *Clostridia* antibodies mimic INSR by binding to INS, *Clostridia* antibodies bind to INSR antibodies. It follows that at least some COX antibodies are not only complementary to *Clostridia* antibodies, as demonstrated in the previous Section, but that this complementarity involves INS- and INSR-specific idiotypes. It was further demonstrated that, as predicted from the similarity data in Section 2.1, PTPN(IA-2) and GAD antibodies could be substituted for INS and *Clostridia* antibodies, yielding the same binding to COX antibodies. The basic logic of these experiments is summarized in Figure 17.

The complementarity of INSR-like COX sequences and INS-like *Clostridia* sequences were demonstrated using U.V. spectroscopy in Section 2.5 and Section 2.6, and it was then shown that TCR derived from T1DM patients recognized these COX and *Clostridia* peptides as antigens, providing direct evidence of autoimmunity against both microbes in three sets of patient TCR. These results indicated that each patient had TCR against both INS–*Clostridia* and against INSR–COX. Similar results were obtained using sera from T1DM patients, which recognized inactivated *Clostridium sporogenes* and COX antibodies (Section 2.7 and Section 2.8), thus demonstrating the existence within the sera of anti-idiotype antibodies against both *Clostridia* antibodies and COX antibodies. Not surprisingly, given the prevalence of the microbes, some of the healthy and T2D control sera also recognized *Clostridium* antigen, *Clostridium* antibodies, and COX antibodies, though generally with lower affinities than the T1DM sera (Figure 15).

In short, *Clostridia* antigens mimic INS; both are in turn mimicked by some INSR antibodies as well as some COX antibodies. COX antigens mimic INSR so that COX antibodies mimic INS as well as INS-like *Clostridia* antigens. Thus, COX antigens are complementary to *Clostridia* antigens; COX antibodies are complementary to (idiotype–anti-idiotype) *Clostridia* antibodies; which means that COX antigens mimic antibodies against *Clostridia* as well as INSR, while *Clostridia* antigens mimic antibodies against COX as well as INS (Figure 17). The result is that the immune system loses the ability to differentiate between “self” and “non-self” because any simultaneous response to both COX and *Clostridia* necessitates an active response against its own antibodies (and TCR) as well INS and INSR. Moreover, each antibody mimics one of the microbial antigens, creating further confusion. A combination of EBV with Staphylococci may induce similar self–non-self confusion, leading to T1DM.

### 3.2. Relationship of the Experimental Findings to Previous Results

The experimental findings reported here are consistent with much of the previously published literature concerning T1DM etiology and pathogenesis. While a variety of viruses and bacteria have been associated with T1DM onset (reviewed in [76]), enteroviruses, and in particular COX, have been the ones most consistently identified through methods ranging from direct virus isolation to antibody cross-reactivity to microbiome studies [12,13,14,15,16,17,18,19,19,20,21,22,23,48,49,50,51,52,53,54,55,56,76,77,78,79,80,81]. Our results also make a strong case of a role for enteroviruses in T1DM etiology but suggest that the main target of enterovirus antibodies is not INS (as most previous research has attempted to demonstrate) but rather INSR, a T1DM target other investigators have previously reported [4,5,6,7,8,9,54,82,83,84,85,86]. Additionally, previous studies have found that COX does not elicit anti-INS antibodies, nor antibodies against GAD or PTPN(IA-2) [38,51,53,54,87,88,89,90,91]. These results are consistent with our previous report that GAD, PTPN(IA-2), and INS share many homologies [73,74] and the results shown in Figure 11 demonstrating that COX antibodies are complementary to, rather than mimics of, GAD and PTPN(IA-2) antibodies.

What is missing from the enterovirus story is how INS becomes a primary target of autoimmunity in T1DM. Given the importance of INS autoantibodies [40,41,42,66,92,93,94,95,96,97,98], as well as anti-GAD and anti-PTPN(IA-2) autoantibodies and TCR [99,100,101,102,103], in T1DM, and the failure of COX antibodies to recognize these antigens, a critical question is whether there needs to be a second microbe that triggers concomitant anti-INS autoimmunity. Our results strongly implicate *Clostridia* for this role.

While few studies have directly linked the presence of a *Clostridium* infection to initiation of T1DM [104,105], a very large number of studies have implicated dysregulation in the number and types of gut *Clostridia.* In particular, *Bifidobacteria, Bacteroides*, and *Lactobacilli* all decrease significantly, while the number of *Clostridia* increases and is directly correlated with the degree of glucose dysregulation observed in the patient [26,37,106,107,108,109,110,111]. Thus, it is plausible that *Clostridia* plays a role in T1DM etiology, either to produce bystander activation of the immune system or, as is more likely in view of the complementarity to COX demonstrated here, by synergizing with COX via the production of complementary immune responses. Microbiome constituents such as *Lactobacilli*, *Bifidobacteria*, and *Bacteroides* may become accidental targets of the resulting autoimmunity because they, too, express antigens that mimic INS (Figure 1 and Figure 2) [112]. Additionally, our results are consistent with the observation that pro-INS and GAD65 are both recognized as antigens by the same T cells [1,38,39,113] so that *Clostridia* could initiate autoimmunity against both antigens via shared mimicry and/or epitope drift.

### 3.3. Animal Models

Two animal models support the theory that T1DM has a multifactorial etiology involving a combination of viral and bacterial infections and, in particular, that *Clostridia* has a role in triggering the disease.

T1DM can be triggered in Lewis rats by infecting them with Kilham rat virus (KRV). Studies found that KRV infection significantly increased the abundance of intestinal *Bifidobacterium* and *Clostridium* species, indicating a possible synergism between the KRV and these bacteria. Furthermore, treating KRV-infected rats with a combination of trimethoprim and sulfamethoxazole (Sulfatrim) beginning on the day of infection prevented the increase in *Bifidobacterium* and *Clostridium* abundance, and also T1DM development [114].

T1DM can also arise “spontaneously” in genetically-predisposed nonobese diabetic (NOD) mice. However, several studies suggest that bacteria, and in particular *Clostridia*, are involved in triggering diabetes. Tanca et al. [115], for example, found that NOD mice, compared with genetically modified NOD mice protected from T1D (Eα16/NOD), differed in the significant depletion of commensal *Clostridial* butyrate biosynthesis species. Consistent with this finding, Jia et al. [64,116] demonstrated that supplementing the gut microbiome of NOD mice with the probiotic *Clostridium butyricum* protected them against diabetes onset. Fecal transplants from non-NOD mice into NOD mice also prevented onset of T1DM, specifically increasing colonization by commensal *Clostridia* species [117]. Conversely, vancomycin-treated NOD mice were much more prone to develop T1DM than non-treated NOD mice, and the accelerated risk was again associated with a significant decrease in commensal *Clostridia* species in the gut providing a niche for pathogenic forms [118].

### 3.4. Epidemiology of Clostridium and Enterovirus Infections

The epidemiology of *Clostridium* and enterovirus infections is also consistent with the possibility that T1DM is a result of their co-infection. Such a multifactorial mechanism helps to explain one of the great mysteries of autoimmune disease epidemiology, and that of T1DM in particular, which is why many of the putative triggers are so common and the incidence of disease so rare.

Overt *Clostridium difficile* infections are relatively common, particularly in children, with an incidence of 4 in 1000 American children or 6 per 10,000 patient days [119]. The rate is approximately half that in most European nations, and adults contract *Clostridia* infections at somewhat lower rates worldwide [120]. However, studies of asymptomatic *C. difficile* carriage demonstrate that approximately 12% of children less than 18 years of age are infected with non-toxigenic variants, while 6% carry toxigenic variants (reviewed in [121]). In most countries, there is no seasonal variation in incidence [122].

There are also approximately 10 million (1 in 35) COX infections each year in the U.S., the majority among infants and children, with similar rates in most other nations [123,124]; however, asymptomatic carriage of enteroviruses as a whole is approximately 5% among children under 18 worldwide [125]. Cases follow cyclical patterns that vary seasonally across the globe. In North America, cases tend to increase sharply during summer and again in late fall, with a peak in August [126], while, for instance in China, the peak is in January–February [127]. Like Clostridia, COX infections also occur in adults at slightly lower rates than in children [123,124,125,126,127].

The epidemiology of COX and *Clostridium* infections broadly correspond to T1DM epidemiology. The incidence of new T1DM diagnoses is approximately equal among individuals less than 19 years of age and in those above 19 years of age [128,129], making children approximately four or five times more likely to develop T1DM than adults in any given year of life. This epidemiology is consistent with the majority of COX and *Clostridia* infections occurring in children. Additionally, the seasonal incidence of new T1DM diagnoses correlates reasonably well with COX incidence. The peak of new T1DM diagnoses in China occurs between December and February [130], which corresponds to the January peak in COX infections. In the U.S., there are two peaks, one of them in August and the other in early winter [130], also corresponding again to the variations in COX incidence. The incidence of new T1DM diagnoses in the southern hemisphere tends to be inverted from that in the northern hemisphere [130], which again corresponds with peak incidences of COX infections [126]. Thus, although a genetic component to T1DM susceptibility is well-recognized among children with relatives with T1DM [131,132], and their risk can be documented by the development of an increasing number and diversity of T1DM-related autoantibodies over many months or years preceding T1DM diagnosis [133,134], the seasonality of new T1DM diagnoses suggest that triggering full-blown autoimmune disease, even against this genetic background, may require an appropriate combination of infectious triggers.

Note, however, that the putative triggers of T1DM identified here—COX with Clostridia—are very common infections, while T1DM is very rare. The reported rates of diagnosed infections and asymptomatic carriage of COX (or enteroviruses more generally) and *Clostridia* raise serious problems for any T1DM mechanism that is based on a mono-infectious trigger model. The estimated number of new, annual T1DM diagnoses worldwide is many orders of magnitude less than the number of new enterovirus and *Clostridia* infections. Estimates of the annual incidence of new T1DM diagnoses in the United States range from approximately 40,000 cases, or 1.2/10,000 individuals [128] to approximately 60,000 cases, or 2.0/10,000 individuals [129], the latter figure being typical of most of the rest of the world [129]. While genetic predisposition certainly accounts for some of T1DM risk [131,132,135], it is important to stress that 90% of new T1DM cases have no known relative with T1DM or any defined genetic risk [136], and known genetic risk factors appear to be involved in fewer new cases each year [137].

The incidence of new T1DM diagnoses is much more in line with a dual, concurrent-infection model of etiology. Assuming 12% of children have *C. difficile* carriage and 5% contract COX each year, then 60/10,000 would be exposed to both in a single year. However, the dual, concurrent-infection model requires that both infections be present simultaneously. Therefore, it seems reasonable to divide the 60/10,000 figure by 52 to yield the probability that an individual would contract both infections during the same week. This yields a probability of 1.2/10,000, which approximates the actual incidence of new T1DM diagnoses. Multiple *Clostridia* species might be involved as triggers of T1DM, so this number might increase, but on the other hand, a requirement for active (or activated) infection with at least one microbe (see the KRV rat model above) might decrease the probability.

Our data also hint at a combination of EBV with *Streptococci* as possible co-triggers of T1DM, which would have a probability of occurring concurrently at approximately the same order of magnitude of occurrence as COX + *Clostridia*. The essential point is that a dual-infection model gets the probability of a new T1DM diagnoses within the right order-of-magnitude estimation, whereas any single-agent model is several orders of magnitude off. Variations in the geographic incidence of T1DM would then be a function of the seasonal variations in the infections and their specific co-incidences in that location.

### 3.5. Prevention Implications of Complementary Antigen Theory

If both COX and *Clostridia* are necessary to trigger T1DM, then several implications concerning the prevention of T1DM follow. One is that anyone diagnosed with either an enterovirus infection or a *Clostridium* infection should be tested for the complementary infection. In light of the fact that T1DM could be prevented in KRV-infected rats with a combination of trimethoprim and sulfamethoxazole (Sulfatrim), beginning on the day of KRV infection [114], if both COX and *Clostridia* infections are present, appropriate antibiotic therapy might be an effective T1DM preventative. The assumption that the clinically obvious infection is the only infection present may be putting patients at risk for post-vaccinal autoimmune complications such as T1DM.

Next, vaccination against COX should be effective in preventing most cases of T1DM. Each individual vaccine should be completely safe in light of Figure 17 because any host cross-reactive epitopes will either be non-antigenic or tolerized. Only in the rare instance that an individual contracts a Clostridium infection concurrently with their COX vaccination would there be a risk of triggering post-vaccinal T1DM (and thus individuals should be screened before vaccination). COX vaccines for this purpose are already under development [138], but these are being developed under the assumption of a mono-infectious etiology for T1DM acting by means of one of three mechanisms: “Beta-cell death may be primarily induced by CVB [COX type B] itself, possibly in the context of poor immune protection, or secondarily provoked by T-cell responses against CVB-infected beta cells. The possible involvement of epitope mimicry mechanisms skewing the physiological anti-viral response toward autoimmunity has also been suggested… Understanding which [mechanisms] are at play is critical to maximize the odds of success of CVB vaccination, and to develop suitable tools to monitor the efficacy of immunization and its intermingling with autoimmune onset or prevention” [138]. If, in fact, T1DM etiology involves complementary antigens, then the safety of COX vaccines may require antigen deletion of regions mimicking INSR and other TIDM autoantigen sequences identified here.

A third approach would be to develop *Clostridium* vaccines, which should be safe for the same reasons provided above for COX vaccines but carry the same caveat regarding COX co-infection. Such *Clostridia* vaccines are also in development, although at present for different purposes [75,139,140,141,142]. The use of such vaccines again depends upon understanding the mechanism by which *Clostridium* is involved in T1DM etiology and including (or excluding) appropriate antigenic regions. For example, our data suggest (but certainly do not prove) that *Clostridium* toxin A is not involved in T1DM pathogenesis so that a toxin-based vaccine might be particularly safe from the perspective of preventing *Clostridium* infections without risk of T1DM as a post-vaccinal side effect. However, if non-toxin-producing strains of *Clostridia* can trigger T1DM, then a toxin-based vaccine may not be optimal for preventing T1DM.

Whether COX or *Clostridia*-based, optimization of T1DM prevention strategies rely, in the end, on having an appropriate animal model with which to test such strategies.

### 3.6. The Need for New Animal Models of T1DM

If T1DM is, in fact, triggered by a combination of COX and *Clostridium* infections (or EBV + *Streptococci*), then new animal models of the disease need to be developed, not least in order to test potential preventative approaches such as those described in the previous section. Such models might be implemented in several ways. One would be to infect susceptible animals with combinations of COX and *Clostridia*. An alternative would be to inoculate animals with combinations of inactivated COX and inactivated *Clostridia*. Success of the second type of experiment would also demonstrate that the pathogenesis is immunologically mediated (and therefore purely autoimmune) rather than requiring damage to the pancreas due to active infection. This second type of experiment might also be used to screen COX and *Clostridia* vaccines for their potential synergy and thus be used as a screen to increase their safety or to warn against their co-administration. Finally, another way to develop a novel T1DM animal model would be to inoculate animals with combinations of COX polyclonal antibodies and *Clostridia* polyclonal antibodies induced in the same species as that inoculated. According to the antigenic complementarity demonstrated in this paper, the resulting immune complexes should mimic the key complementary antigens triggering T1DM, and their idiotypes should therefore function equally as antigenic epitopes to initiate an autoimmune process.

Animal models should also explore the possibility suggested by the data we have provided here that an EBV–*Streptococci* combination may trigger T1DM.

### 3.7. Limitations of the Study

This study has limitations. One obvious one that has just been addressed is the lack of an animal model to test whether a combination of COX and *Clostridia* induces T1DM. The previous section lays out three ways to test this prediction.

A second limitation is that we tested our human sera only for binding to *C. sporogenes* rather than pathogenic species of *Clostridia* such as *C. perfringens, C. difficile*, etc., and may therefore have missed important cross-reactivity to additional *Clostridia* antigens or found cross-reactivities that do not extrapolate to other *Clostridia* species. Similarly, we were unable to obtain even relatively pure whole-COX antigens and therefore could not directly demonstrate whether our human T1DM sera were positive for COX antibody. COX antibody presence was inferred from a demonstration of anti-idiotypic responses to *Clostridia* antibodies combined with a demonstration of COX–*Clostridia* antibody anti-idiotype.

A third limitation is that the actual molecular complementarity between COX and *Clostridium* antigens has only been tested in the most cursory way in the present study using two pairs of peptides. Clearly much additional work needs to be performed to characterize this antigenic complementarity.

Similarly, the role of TCR in mediating T1DM through both antigenic complementarity and TCR idiotype–anti-idiotype interactions has only been explored cursorily here. While much more comprehensive studies have previously been published [72,73], the role of TCR complementarity needs much further investigation. The possibility that the TCR sequences expanded in T1DM may identify not just the particular microbes triggering the disease but also the specific antigenic sequences that are involved also needs further research. Both of these possibilities therefore stand as important predictions that can be used to test the validity and utility of the complementary antigen theory presented here.

Additionally, the COX–*Clostridia* combination and its corresponding anti-idiotype may not be the sole triggers of T1DM. Our data also suggest that an EBV–*Streptococci* combination be investigated more thoroughly. Additionally, *Bifidobacteria* and *Lactobacilli* both appeared as possible INS mimics in our similarity studies (Figure 1 and Figure 2) and have also been implicated in many T1DM microbiome studies (Section 3.2 and Section 3.3). Unfortunately, no antibodies against these bacteria could be located and therefore their possible synergy with COX or other viruses could not be explored.

Correspondingly, various viruses other than COX and EBV have also been associated with T1DM initiation, including CMV [89,143,144,145] and rotaviruses [145,146]. Thus, it is possible, if not likely, that other microbial combinations (probably bacterial–viral but possibly bacterial–bacterial or viral–viral) can also express complementary antigens that mimic human glucose-regulatory proteins and peptides and thus induce some form of T1DM. This possibility is supported by the fact that not all of the T1DM sera tested in this study reacted strongly to *Clostridia* antigens (Section 2.7) and some cross-reactivity was observed between non-COX virus antibodies or bacteria to INSR peptides (Section 2.2). Thus, it is important not to over-interpret the data presented here as meaning that a COX–*Clostridium* combination is the only possible cause of T1DM or predict that COX and/or *Clostridium* vaccines will prevent all T1DM cases in the future.

Finally, there is thus far no direct evidence from human studies (either clinical or epidemiological) demonstrating that individuals newly diagnosed with T1DM have recently been exposed to both enteroviruses and *Clostridia* infections. Such studies will be needed to test the hypothesis proposed here.

## 4. Materials and Methods

### 4.1. Similarity Searches

Protein sequences for use in similarity searches were obtained from UniProtKB (https://www.uniprot.org/help/uniprotkb) accessed between 28 August 2018 and 17 July 2022. BLASTP (version 2.2.31+) on the www.expasy.org server. BLOSUM80 (1 June 2021–17 July 2022) was used to identify the type of short, continuous sequences approximately ten to fifteen amino acids in length that are presented by Human Leukocyte Antigens (HLA) to T and B cells [147,148]. The E value was set to 1; filter low complexity regions on; no gaps; 100 best scoring and best alignments to show. Only matches that had a Waterman–Eggert score of at least 50, an E value of less than 1.0, and which contained a sequence of ten amino acids in which at least six were identical, were counted as sufficiently similar to induce possible cross-reactive immunity; this criterion is based on substantial experimental research demonstrating that sequences exhibiting at least this degree of similarity have a >85% probability of being cross-reactive under experimental conditions [149,150,151,152].

### 4.2. Experimental Protocols

ELISA and double-antibody ELISA (DA-ELISA) were employed to investigate whether the similarity searches yielded immunologically valuable information.

Enzyme-linked immunosorbent assay (ELISA) was used to investigate cross-reactivities between microbial antibodies and diabetes-related proteins. The diabetes-related protein was diluted in pH 7.4 phosphate buffer to a concentration of 10 µM. This standard solution was then diluted by ten-fold steps to approximately 10^−14^ M. Two wells received only phosphate buffer as controls. An amount of 100 µL of each protein dilution was added in duplicate to wells of a Costar round-bottomed 96-well ELISA plate and incubated for one hour. The excess protein was triply washed out using a 1% Tween 20 solution (in phosphate buffer) and a plate washer. Next, 200 µL of blocking agent (2% polyvinylalcohol in phosphate buffer) was added to every well, incubated for an hour, and then triply washed. (PVA was used rather than bovine serum albumin or ovalbumin because these proteins were found to cross-react with some of the antibodies used in our experiments.) An antibody against a microbe (at 1 mg/mL concentration) was then diluted to 1/200 in phosphate buffer, and 100 µL was added to every well. The antibody was incubated for an hour and then triply washed. A species-appropriate horseradish peroxidase-linked secondary antibody was then, at a dilution of 1/1000, incubated for an hour, and triply washed. Finally, 100 µL of ABTS reagent (Chemicon via SigmAldrich, St. Louis, MO, USA) was added, incubated for 30 min, and the plate read at 405 nm in a Spectramax UV-VIS scanning spectrophotometer (Molecular Devices, San Jose, CA, USA) Data were gathered using Spectramax software (Molecular Devices, San Jose, CA, USA) and then analyzed using Excel (Microsoft, San Jose, CA, USA). Analysis essentially consisted of subtracting non-specific binding to the buffer-only wells from the protein-containing wells and plotting the amount of antibody binding (as measured by absorbance at 405 nm) as a function of protein concentration.

Double antibody ELISA (DA-ELISA) was used to investigate possible antigenic complementarity between the antibodies used in the study. DA-ELISA differs from ELISA in that the protein laid down in the 96-well plate in the initial step of an ELISA is substituted with an antibody. A second antibody (from a different species) is tested for its ability to bind to the first. The ability of the second antibody to bind to the first is then monitored using peroxidase-linked antibody against the species from which the second antibody is derived. As in the ELISA protocol, the first antibody is made up at a concentration of approximately 10 µM (assuming IgG antibodies have a molecular weight of 150,000 daltons) and then serially diluted by factors of ten. The rest of the protocol is the same. In some cases, it is possible to use an HRP-conjugated antibody as the second antibody in the process, in which case the third step can be skipped. The use of HRP-conjugated antibodies in the second step permits the first and second antibodies to be from the same species.

### 4.3. Antigens

INS receptor peptides and glucagon receptor peptides were synthesized to at least 95% purity by mass spectrometry by RS Synthesis (St. Louis, MO, USA). See Figure 8 and Figure 9 for sequences. Inactivated *Clostridium sporogenes* was obtained from Professors Moumita Rakshit and Swee Hin Teoh of Nanyang Technological University, Singapore, where it is being developed by GN Neutriceuticals SDN BHD in Malaysia as a possible cancer therapy [75]. Proteins and peptides are listed in Table 2.

### 4.4. Antibodies

Antibodies against viruses are listed in Table 3; against bacteria in Table 4; against proteins in Table 5; and secondary antibodies are listed in Table 6.

Monkey COX B4 was observed using anti-human secondary antibody.

### 4.5. Human Sera

Human sera sources are listed in Table 7.

## Figures and Tables

**Figure 1 ijms-24-08336-f001:**
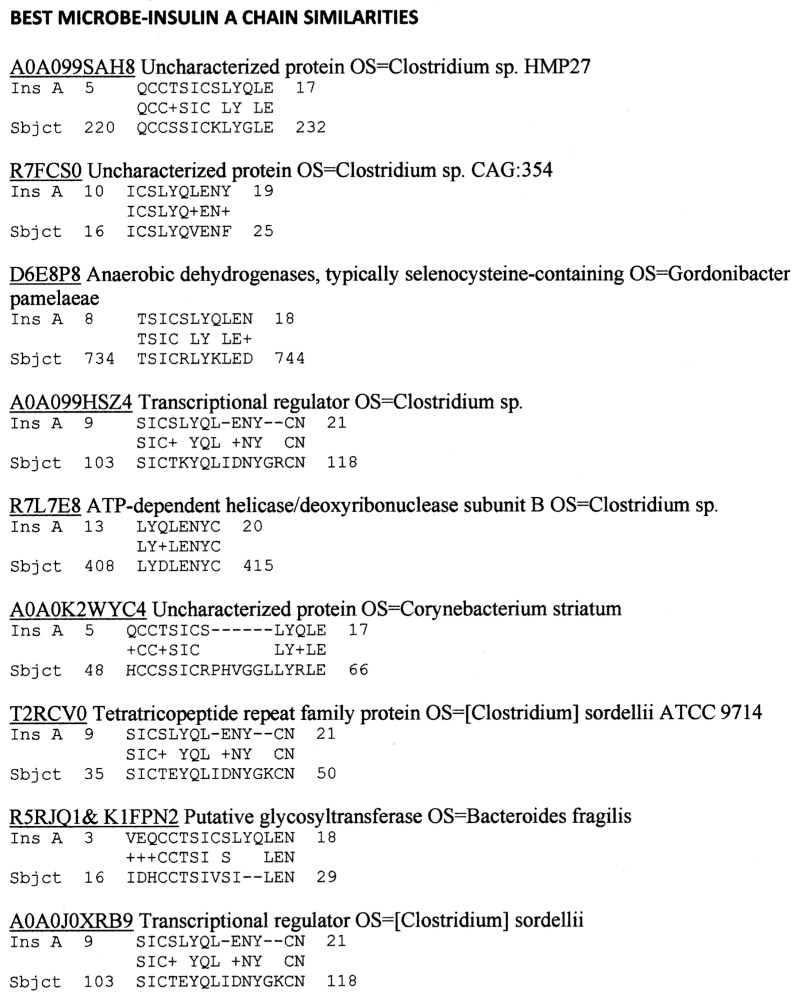
Results of BLASTP similarity search comparing the human INS A chain sequence with the UNIPROTKB bacterial database curated for human pathogens and commensals.

**Figure 2 ijms-24-08336-f002:**
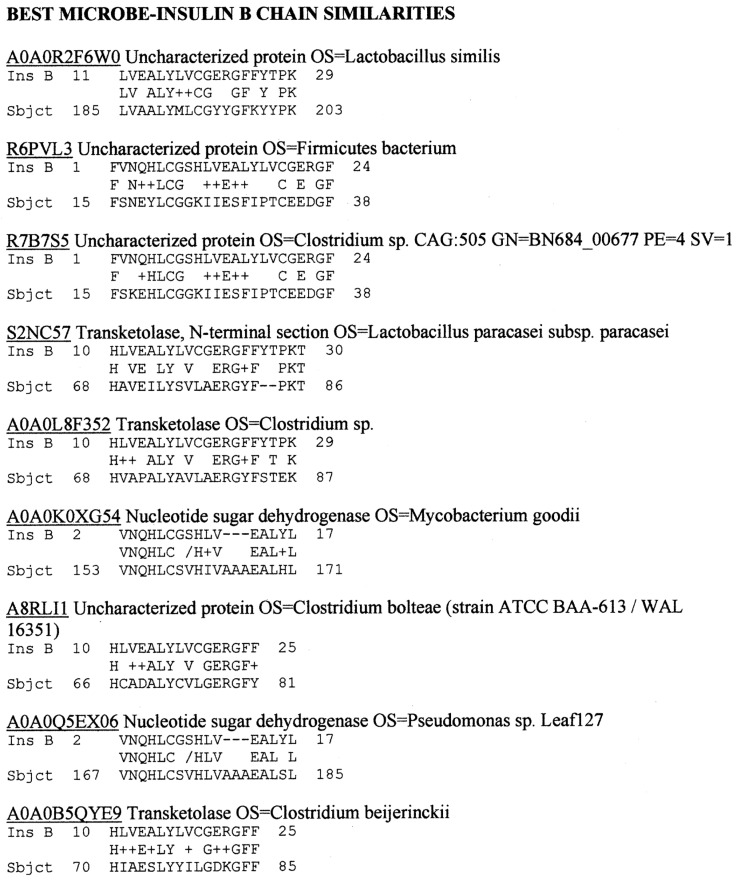
Results of BLASTP similarity search comparing the human INS B chain sequence with the UNIPROTKB bacterial database curated for human pathogens and commensals.

**Figure 3 ijms-24-08336-f003:**
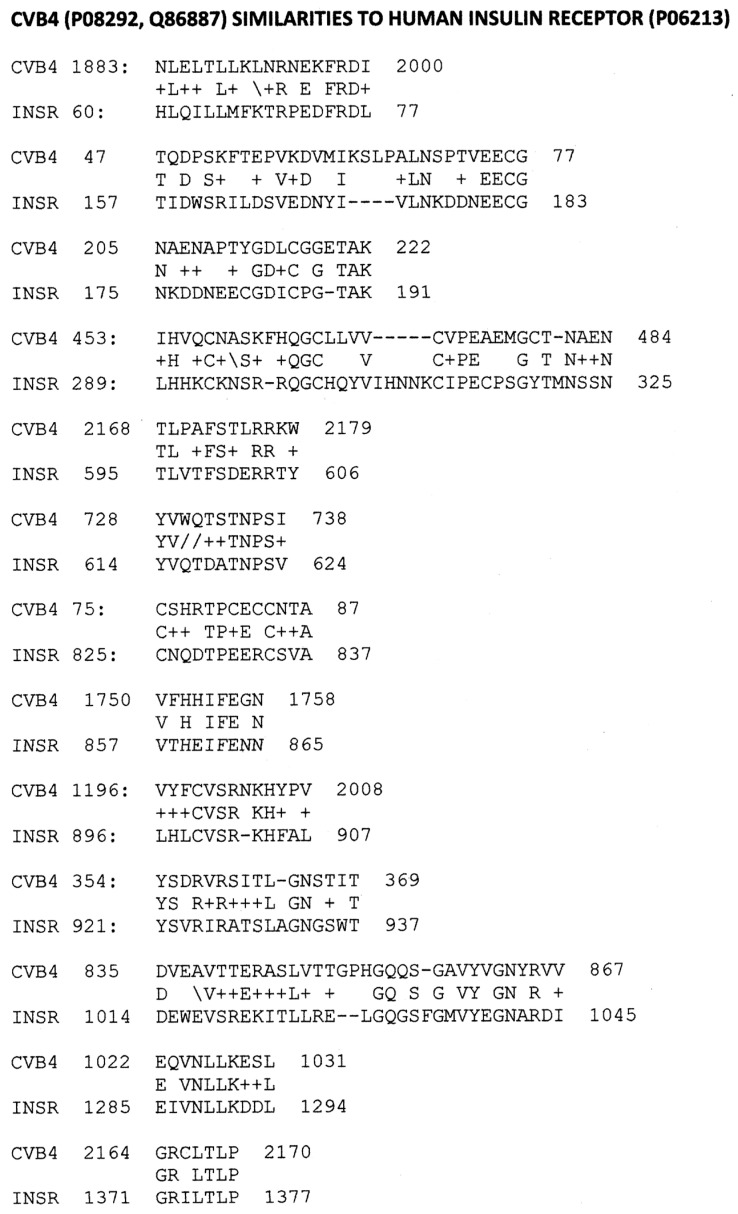
BLASTP similarities between COX type B4 and the human INS receptor.

**Figure 4 ijms-24-08336-f004:**
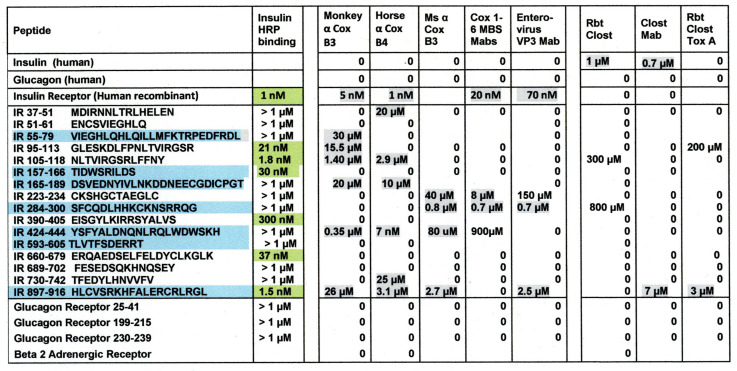
Summary of quantitative ELISA experiments regarding COX and Clostridia antibody binding to INS, glucagon, the INS receptor (IR), IR peptides, glucagon receptor peptides, and the beta 2 adrenergic receptor. Blue highlighted sequences are ones overlapping the COX-INS receptor similarities listed in Figure 3. Green highlighted figures are sequences with significant INS binding. Gray highlighted figures are sequences that displayed significant antibody binding. Results are binding constants (Kd) in micromoles determined by the inflection points of each curve. A zero (0) indicates that binding was not measurable (in practice, Kd > 1 micromolar). The IR sequences are listed by the numerical position in the IR sequence and followed by the single-letter amino acid sequence. HRP = horseradish peroxidase conjugated; α = “anti-“; Cox = coxsackivierus; Mabs = monoclonal antibodies (mouse); Rab = rabbit antibody; Clost = *Clostridium*; Tox A = *Clostridium* toxin A.

**Figure 5 ijms-24-08336-f005:**
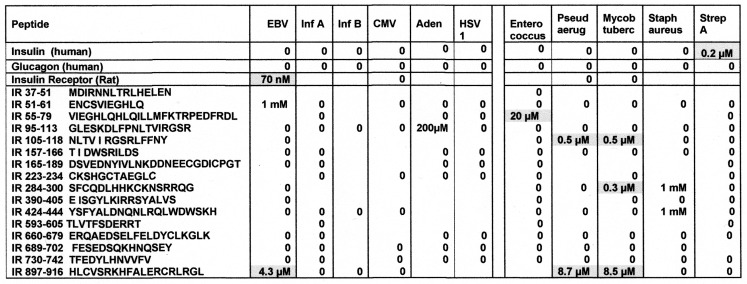
Binding of virus and bacterium antibodies to INS, glucagon, the INS receptor (INSR), and INSR peptides. EBV = Epstein–Barr virus; Inf A = Influenza A virus; Inf B = Influenza B virus; CMV = cytomegalovirus; Adeno = adenovirus; HSV1 = human herpes simplex virus type 1; Pseud aerug = *Pseudomonas aeruginosa*; Mycob tuber *= Mycobacterium tuberculosis*; Staph aureus = *Staphylococcus aureus*; Strep A = group A *Stretpcococci*. See Figure 8 for an explanation of the rest of the abbreviations and formalisms.

**Figure 6 ijms-24-08336-f006:**
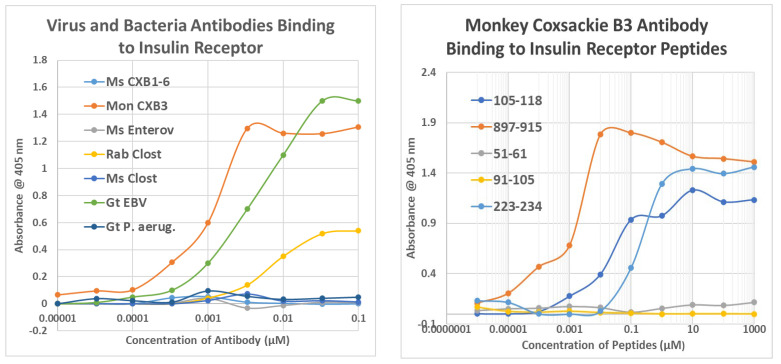
Results of ELISA studies concerning binding to the INS receptor (IR) and its peptides. LEFT: Monkey (Mon) anti-coxsackievirus type B3 (CXB3), Epstein–Barr virus (EBV) antibody and to a much lesser extent, *Clostridium* antibodies, bind to the human insulin receptor. RIGHT: Further characterization of monkey anti-CXB3 antibody binding to peptides derived from the human INSR. Ms = mouse; Rab = rabbit; CX = coxsackievirus; Enterov = enterovirus; EBV = Epstein–Barr virus; Clost = *Clostridium; P. aerug. = Pseudomonas aeruginosa.* Numbers indicate the amino acid sequence of the INS receptor from which the peptides were derived. Further data are provided in Appendix D.

**Figure 7 ijms-24-08336-f007:**
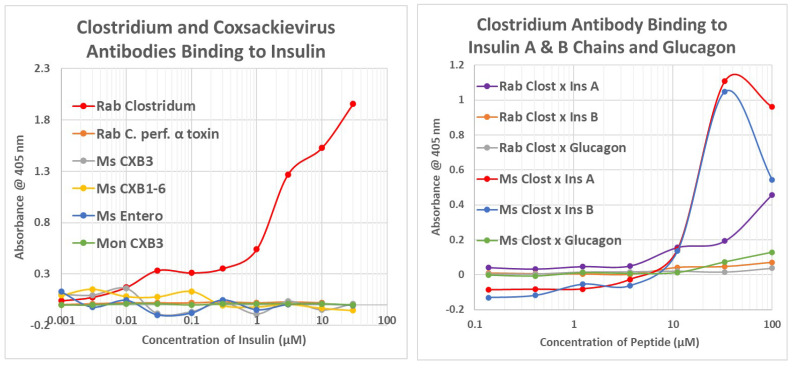
Results of ELISA studies concerning binding to insulin (INS). LEFT: Clostridium antibody, but not coxsackievirus (X) antibodies, bind to INS. RIGHT: *Clostridium* antibodies bind to the INS A and B chains but not to glucagon. Rab = rabbit; Ms = mouse; C. perf toxin = *Clostridium perfringens* alpha toxin; Cox = COX; Clost = *Clostridium*; Ins = INS; A = A chain; B = B chain.

**Figure 9 ijms-24-08336-f009:**
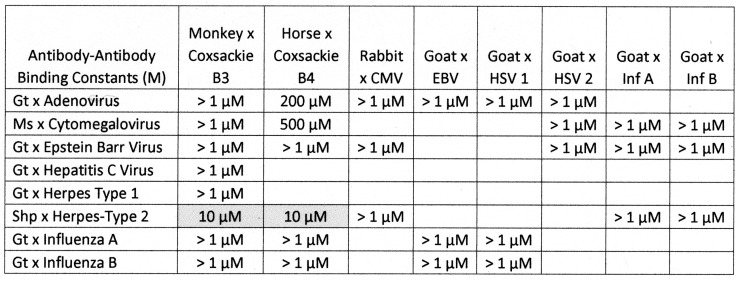
Summary of the binding constants (Kd) from double-antibody ELISA experiments involving a virus antibody potentially binding to another virus antibody. See Figure 14, Figure 15 and Figure 16 for examples. COXB = COX type B; CMV = cytomegalovirus; HBV = hepatitis B virus; HCOX = hepatitis C virus; HSV = herpes simplex virus; Ms = mouse; Gt = goat; Shp = sheep.

**Figure 10 ijms-24-08336-f010:**
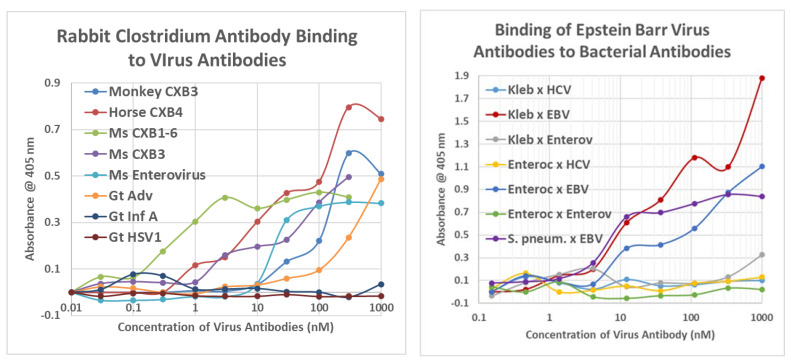
Representative binding curves resulting from double antibody ELISA experiments involving virus and bacterium antibodies. LEFT: Rabbit *Clostridium* antibodies bound to all of the COX antibodies tested as well as an enterovirus antibody but did not bind to other virus antibodies. RIGHT: Additional control combinations of virus–bacterial antibody pairs tested by double antibody ELISA and summarized in Appendix E. CXB = Coxsackievirus; Adv = adenovirus; Inf A = influenza A virus, HSV1 = herpes simplex type 1 virus; Ms = mouse; Gt = goat; Hs = horse; Mon = monkey; Enteroc = *Enterococcus faecium*; *S. pneum. = Streptococcus pneumoniae*; Adv = adenovirus; EBV = Epstein–Barr virus; HCOX = hepatitis C virus; Enterov = enterovirus. See Figure 8 and Figure 9 for species of each antibody.

**Figure 11 ijms-24-08336-f011:**
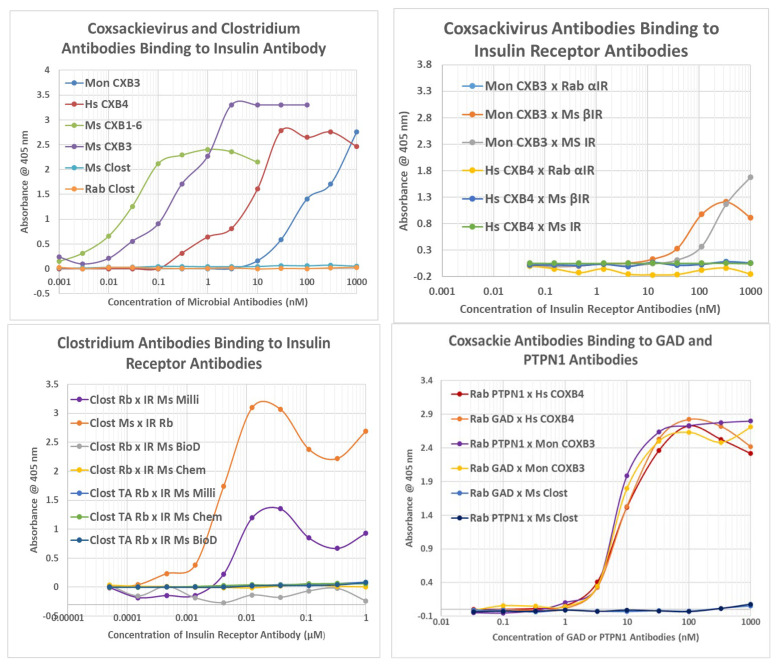
Results of double antibody ELISA experiments. TOP LEFT: Coxsackievirus (CX) and Clostridium antibodies binding to INS antibody. TOP RIGHT: CX and Clostridium antibodies binding to several INS receptor antibodies. BOTTOM LEFT: *Clostridia* antibodies binding to INS receptor antibodies (IR-Ab). Rb IR-Ab were tested against Ms *Clostridia*, while Ms IR-Ab were tested against Rb *Clostridia.* BOTTOM RIGHT: Binding of COX type B (COXB) or *Clostridium* (Clost) antibody binding to glutamic acid decarboxylase (GAD-65) antibody or protein tyrosine phosphatase non-receptor type 1 (PTPN(IA2)) antibody. Clost = Clostridium; TA = toxin A; Rb = rabbit; Ms = mouse; Milli = Millipore CXB = Coxsackievirus type B; IR = INS receptor (alpha and beta indicating the chain to which the antibody is specific); Mon = monkey; Hs = horse; Rab = rabbit; Ms = mouse; Biod = Biodesign; Chem = Chemicon (see Tables of antibodies in Section 4).

**Figure 12 ijms-24-08336-f012:**
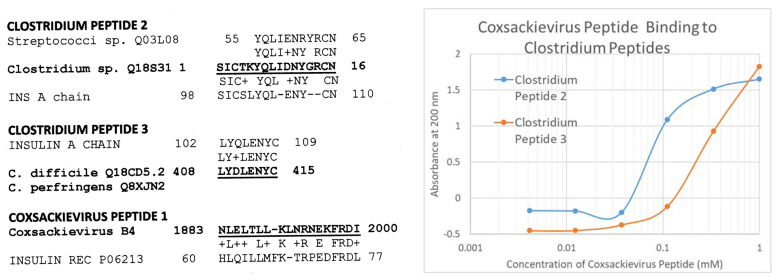
Two *Clostidium* peptides that mimic the INS A chain (INS A) and one COX peptide that mimics the INS receptor (INS Rec) were synthesized (sequences and similarities to the LEFT) and tested for their ability to bind to each other using ultraviolet spectrometry (RIGHT). Kd was 60 µM for *Clostridium* peptide 2 and 200 µM for peptide 3.

**Figure 13 ijms-24-08336-f013:**
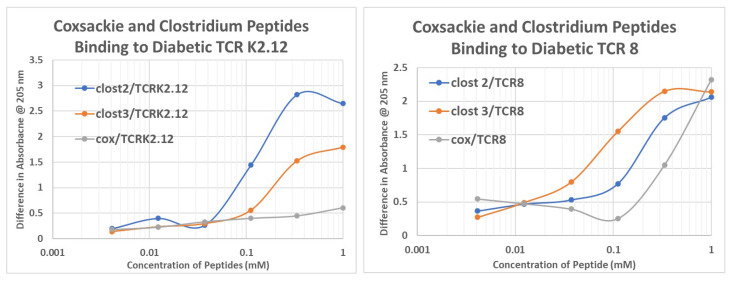
Ultraviolet spectrometry study of COX (cox) peptide (which mimics the INS receptor) and *Clostridium* peptides 2 and 3 (clost2, clost3) (which mimic INS), binding to T cell receptors (TCR) K2.12 (which mimics the INS receptor) and TCR8 (which mimics the glucagon receptor). The binding constants listed in Table 1 were estimated from the inflection points of the curves.

**Figure 17 ijms-24-08336-f017:**
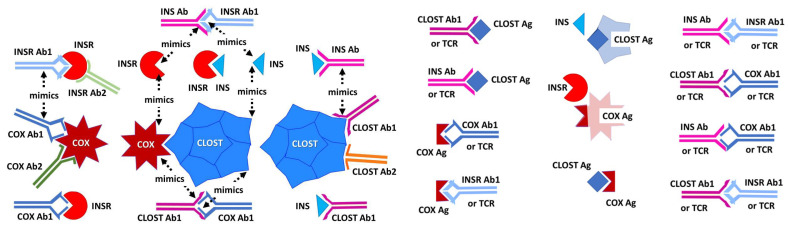
Summary of the various types of complementarity and mimicry revealed by the experiments performed here. INS = insulin; INSR = INS receptor; Ab = antibody; TCR = T cell receptor; Clost = *Clostridium*; COX = coxsackievirus. The arrow link mimics. Steric fits represent complementarity between antibodies with their antigens as well as between pairs of antigens (e.g., COX and Clost or INS and INSR).

**Table 1 ijms-24-08336-t001:** Summary of ultraviolet spectrometry experiments concerning binding of T cell receptor sequences expanded in type 1 diabetic patients (TCR DIA) to a COX peptide (Cox Pep) that mimics the INS receptor (Ins Rec), two *Clostridium* peptides (Clost Pep) that mimic INS, two INS receptor peptides identified by their amino acid sequence numbers, and INS itself. Gluc = glucagon-like sequence; GR = glucagon receptor-like sequence; IR = INS receptor-like sequence; Ins = INS-like sequence. Some TCR sequences mimic regions of more than one protein.

Binding Constants (µM)	Cox Pep #1	Ins Rec 105–118	Ins Rec 897–915	Clost Pep #2	Clost Pep #3	INS
TCR DIA 1 (Gluc) CASSIYLCSVEATRAD	400	125	300	100	100	80
TCR DIA 2, K2.16 (GR) CASSLAVIRT	1000	>1000	>1000	110	>10,000	>1000
TCR DIA 4, K2.4 (IR) CASSLATSGGGSDTQ	1000	>1000	>1000	130	70	15
TCR DIA 7 (IR) CASSFRRVTDTQ	>10,000	>1000	>1000	120	100	70
TCR DIA 8 (Ins, GR) CASSQVRLAGGGEQ	750	120	130	300	70	130
TCR DIA 9 (Ins, GR) CASSLGQGETEAFF	100	150	145	500	100	23
TCR DIA 10 (IR) CASRNLGLNTE	2000	110	140	1000	80	130
TCR DIA 4,8,9 (Ins, IR, GR) DSALYLCASSLG	200	>1000	>1000	3000	150	>1000
TCR DIA K2.12 (IR) CASSDRLGNQPQH	>10,000	120	>1000	100	200	75

**Table 2 ijms-24-08336-t002:** List of proteins and peptides used in the research and their suppliers.

PRODUCT NAME	SPECIES	SUPPLIER	PRODUCT #	Purity
Insulin (INS)	Human (recombinant)	Sigma-Aldrich	I2643	100%
INS A chain, oxidized	Bovine	Sigma-Aldrich	I1633	>80%
INS B chain, oxidized	Bovine	Sigma-Aldrich	I1764	>95%
INS C chain	Human (recombinant)	Sigma-Aldrich	C9781	>95%
INS Receptor	Human (recombinant)	R&D Systems	1544-IR	>95%
Glucagon	Human (recombinant)	Sigma-Aldrich	G2044	100%
Beta 2 Adrenergic Receptor	Rat (recombinant)	MyBioScource	MBS7111177	>90%

**Table 3 ijms-24-08336-t003:** List of primary antibodies against viruses used in this study and their suppliers.

VIRUS ANTIBODIES	SPECIES	SUPPLIER	PRODUCT #
Adenovirus	Goat	Millipore	AB1056
COX B3	Monkey	ATCC (NIH Reference Reagent)	VO31-501-563
COX B4	Horse	ATCC (NIH Reference Reagent)	VO30-501-560
COX B3	Mouse	Millipore	MAB948
Coxsackie Virus B1-B6 Blend	Mouse	Millipore	MAB9410
Cytomegalovirus	Goat	Biodesign International	B562756
Cytomegalovirus	Mouse	Biodesign International	C65861M
Enterovirus pan VP3	Mouse	MyBioSource	MBS319564
Epstein-Barr Virus	Rabbit	Invitrogen	PA5-115471
Hepatitis C Virus core Antigen	Rabbit	Invitrogen	PA1-4113
Herpes Simplex Virus Type 1	Goat	Invitrogen	PA1-7493
Herpes Simplex Virus Type 1/2	Rabbit	Invitrogen	PA1-7214
Influenza A HRP	Goat	Biodesign International	B65243G
Influenza B HRP	Rabbit	Biodesign International	B653446

**Table 4 ijms-24-08336-t004:** List of primary bacteria antibodies used in this study and their suppliers.

BACTERIA ANTIBODIES	SPECIES	SUPPLIER	PRODUCT #
*Clostridia*	Rabbit	Invitrogen	PA1-7210
*Clostridium* sp. HRP	Rabbit	US Biological	C5853-25C
*Clostridium* alpha toxin-HRP	Rabbit	Bioss	Bs-2273R-HRP
*Enterococcus* HRP	Rabbit	Invitrogen	PA1-73122
*Escherichia coli*	Goat	abcam	AB13627
*Klebsiella pneumoniae* HRP	Rabbit	Invitrogen	PA1-73176
*Mycobacterium tuberculosis*	Rabbit	ABD Serotec	OBT0947
*Mycobacterium tuberculosis*	Guinea Pig	MyBioSource	MBS315001
*Pseudomonas aeruginosa*	Guinea Pig	Biodesign International	B47578P
*Staphylococcus aureus*	Rabbit	Invitrogen	PA1-7246
*Staphylococcus aureus* HRP	Rabbit	Invitrogen	PA1-73173
*Streptococcus* Group A	Goat	Invitrogen	PA1-7249
*Streptococcus* Group A HRP	Rabbit	Acris Antibodies	BP2026HRP
*Streptococcus pneumoniae*	Rabbit	Biodesign International	B65831R
*Streptococcus pneumoniae*	Rabbit	Invitrogen	PA1-7259

**Table 5 ijms-24-08336-t005:** List of protein antibodies used in this study and their suppliers.

PROTEIN ANTIBODIES	SPECIES	SUPPLIER	PRODUCT #
Glutamic acid decarboxylase 65	Rabbit	Millipore	ABN101
Insulin (INS)	Rabbit	Sigma	HPA004932
INS Receptor	Mouse	Chemicon	MAB105
INS Receptor alpha	Rabbit	Biodesign International	K54244R
INS Receptor alpha	Mouse	Biodesign International	K54241M
INS Receptor beta	Mouse	Millipore	05-1104
PTPN(IA-2)	Rabbit	Sigma	HPA007179

**Table 6 ijms-24-08336-t006:** List of secondary horseradish peroxidase-labelled (HRP) antibodies used in this study and their suppliers.

SECONDARY ANTIBODIES	SPECIES	SUPPLIER	PRODUCT #
Anti-Guinea Pig-HRP	Rabbit	abcam	AB6771
Anti-Horse IgG-HRP	Goat	Santa Cruze Biotechnology	SC-2448
Anti-Human IgG-HRP	Goat	Sigma	AO170
Anti-Mouse IgG-HRP	Goat	Sigma-Aldrich	A9917
Anti-Rabbit IgGHRP	Goat	Invitrogen	65-6120

**Table 7 ijms-24-08336-t007:** List of human sera used in this study and their suppliers.

Disease State	Supplier	Our ID	Sample ID	Age	Sex	Race	HbA1c %	Serum Treatment
T1DM	Lee Biosolutions	LEE 01	09E5731 A1c-50.11	39	M	Caucasian	11.5	EDTA
T1DM	Lee Biosolutions	LEE 02	09E5731 A1c-50.12	36	F	Caucasian	14.4	EDTA
T1DM	Lee Biosolutions	LEE 03	09E5731 A1c-03	29	F	Caucasian	10.3	EDTA
T1DM	Lee Biosolutions	LEE 04	09E5731 A1c-04	30	M	Caucasian	10.5	EDTA
T1DM	Lee Biosolutions	LEE 05	09E5731 A1c-05	24	F	Caucasian	11.3	EDTA
T1DM	Innovative Resesarch	T1DM #1	HMN889069	61	M	Caucasian		Off the clot
T1DM	Innovative Resesarch	T1DM #2	HMN889070	36	M	Asian		Off the clot
T1DM	Innovative Resesarch	T1DM #3	HMN889071	56	F	Caucasian		Off the clot
T1DM	Innovative Resesarch	T1DM #4	HMN889090	58	M	Hispanic		Off the clot
T1DM	Innovative Resesarch	T1DM #5	HMN889091	49	F	African American		Off the clot
T2D	ZenBio	#69	SER-DPLE2ML		M	Caucasian		Off the clot
T2D	ZenBio	#71	SER-DPLE2ML		F	African Amderican		Off the clot
Healthy	Zen-Bio	Healthy 1	HSER-2ML		F	Caucasian		Off the clot
Healthy	Innovative Resesarch	Healthy 86	39521-086	46	M	African American		Off the clot
Healthy	Innovative Resesarch	Healthy 87	39521-087	36	F	African American		Off the clot
Healthy	Innovative Resesarch	Healthy 88	39521-088	40	M	African American		Off the clot
Healthy	Innovative Resesarch	Healthy 89	39521-089	59	M	African American		Off the clot

## Data Availability

The majority of the data compiled for this study is presented herein; all data not explicitly described are available on request from the primary investigator.

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
