# Peer review of "Clostridia and Enteroviruses as Synergistic Triggers of Type 1 Diabetes Mellitus"

_ijms, 2023, doi:10.3390/ijms24098336_

Round 1

Reviewer 1 Report

The manuscript by Root-Bernstein and co-workers investigate the potential complementarity and mimicry between antibodies against clostridium and coxsackieviruses with INS and INSR in the development of T1DM. The proposed idiotype-anti-idiotype pairs are very interesting, but the paper does not provide any experimental evidence that these are indeed involved in the onset of T1DM. Yet, the discussion on this is very elaborate and I find it too speculative considering the presented results. In general, the manuscript has a vast number of figures which can be reduced considerably. Moreover, the figures do not show any statistics. Taken together, I would suggest a major revision to deal with these issues.

Specific comments:

I find figures 1-8 not very informative. Yes, they show the similarities between the INS and the microbial peptides, but that is not really the main point here. The authors aim to demonstrate that clostridium has the highest number of similarities for INS and that COX has peptides more similar to INSR. Moreover, the shown alignments are just a subset of the significant similarities. I would suggest to find a different way of quantifying the number of similarities and have this summarized in one graph.

Moreover, a general search was done for INS in figure 1 and 2 which showed that clostridium popped up several times in the results. Why was this not done for COX?

In figure 9 fragments of insulin receptor were tested for binding to antibodies to clostridium and COX. Why are these fragments different from the homologous sequences identified in figure 5?

Do Figures 11-13 really contribute to what is already shown in figure 9 and 10? Can’t this be moved to supplemental?

As similar approach may be taken for figures 14-18.

Figure 19 to 21 could be combined to one figure.

The experiments in section 2.7 only involve a very limited number of patients, moreover each figure appears to have a different set of patients.

Figure 26 does not show T2 diabetics although the legend indicates this.

Many traces do not have error-bars and I don’t see any statistics to support significant findings.

The idea that Clostridia antibodies will be complementary to COX antibodies and will form idiotype-anti-idiotype pairs that “confuse” the immune system is an interesting hypothesis to explain why perhaps two infections are needed to develop autoimmunity. How would this confusion work? The authors speculate elaborately on this in the discussion and I find this too speculative in the absence of any experimental work proving that a coinfection of clostridium and COX is indeed sufficient to induce T1DM. I think that the discussion can be significantly reduced in size.

Author Response

Reviewer 1

Open Review

Quality of English Language

( ) English very difficult to understand/incomprehensible
( ) Extensive editing of English language and style required
( ) Moderate English changes required
(x) English language and style are fine/minor spell check required
( ) I am not qualified to assess the quality of English in this paper

Yes

Can be improved

Must be improved

Not applicable

Does the introduction provide sufficient background and include all relevant references?

(x)

( )

( )

( )

Are all the cited references relevant to the research?

(x)

( )

( )

( )

Is the research design appropriate?

(x)

( )

( )

( )

Are the methods adequately described?

(x)

( )

( )

( )

Are the results clearly presented?

( )

(x)

( )

( )

Are the conclusions supported by the results?

( )

( )

(x)

( )

Comments and Suggestions for Authors

The manuscript by Root-Bernstein and co-workers investigate the potential complementarity and mimicry between antibodies against clostridium and coxsackieviruses with INS and INSR in the development of T1DM. The proposed idiotype-anti-idiotype pairs are very interesting, but the paper does not provide any experimental evidence that these are indeed involved in the onset of T1DM. Yet, the discussion on this is very elaborate and I find it too speculative considering the presented results. In general, the manuscript has a vast number of figures which can be reduced considerably. Moreover, the figures do not show any statistics. Taken together, I would suggest a major revision to deal with these issues.

Specific comments:

I find figures 1-8 not very informative. Yes, they show the similarities between the INS and the microbial peptides, but that is not really the main point here. The authors aim to demonstrate that clostridium has the highest number of similarities for INS and that COX has peptides more similar to INSR. Moreover, the shown alignments are just a subset of the significant similarities. I would suggest to find a different way of quantifying the number of similarities and have this summarized in one graph.

WE BELIEVE THAT THE REVIEWER HAS MISUNDERSTOOD THE POINT OF THE SIMILARITY SEARCHES. QUANTITY IS IRRELEVANT.  AS WE AND OTHERS HAVE DEMONSTRATED IN PREVIOUS PAPERS (E.G., doi: 10.1002/bies.201600083. Epub 2016 Sep 5. PMID: 27594308; PMCID: PMC7161894 and  doi: 10.1002/bies.201800117. Epub 2018 Sep 28. PMID: 30264468. ), COMMENSAL MICROBES ARE FAR MORE LIKELY TO DISPLAY STATISTICALLY GREATER NUMBERS OF SIMILARITIES TO HUMAN PROTEINS THAN ARE PATHOGENIC MICROBES. THUS, COMMENSAL FORMS OF E. COLI, MYCOBACTERIA, BIFIDOBACTERIA, LACTOBACILLI, ETC. WILL ALL YIELD GREATER NUMBERS OF SIMILARITIES TO INSULIN OR THE INSULIN RECEPTOR THAN WILL PATHOGENIC BACTERIA. HOWEVER, SINCE THESE BACTERIA ARE TOLERATED BY THE IMMUNE SYSTEM, THOSE SIMILARITIES HAVE NO MEANING FOR TRIGGERING AUTOIMMUNITY. WHAT MAKES CLOSTRIDIA STAND OUT IS THAT THEY ARE THE ONLY PATHOGENIC BACTERIA THAT CONSISTENTLY APPEARED AMONG THE BEST MATCH SIMILARITIES.

IN ANY EVENT, THE PURPOSE OF THE SIMILARITY SEARCHES WAS NOT TO PROVIDE EVIDENCE THAT CLOSTRIDIA ARE CAUSATIVE AGENTS IN T1DM, BUT RATHER TO PROVIDE A NOVEL LEAD UPON WHICH TO BASE EXPERIMENTS. AS THE EXPERIMENTS DEMONSTRATE, OTHER VIRUSES (E.G., EBV) AND BACTERIA (E.G., STREPTOCOCCI) MAY ALSO PLAY ROLES IN TRIGGERING T1DM.

Moreover, a general search was done for INS in figure 1 and 2 which showed that clostridium popped up several times in the results. Why was this not done for COX?

A SIMILAR SEARCH WAS DONE FOR COX BUT NOT REPORTED IN LIGHT OF THE WELL-DOCUMENTED RELATIONSHIP OF COX TO T1DM. IT IS NOW PROVIDED IN APPENDIX 2 AND APPROPRIATE TEXT INSERTED IN SECTION 2.1. AGAIN, THE ISSUE OF COMMENSAL VERSUS PATHOGENIC APPLIES.

In figure 9 fragments of insulin receptor were tested for binding to antibodies to clostridium and COX. Why are these fragments different from the homologous sequences identified in figure 5?

THE SYNTHESIS OF SEVERAL OF THE HOMOLOGOUS SEQUENCES COULD NOT BE COMPLETED BEFORE THE DUE DATE FOR THE SUBMISSION OF THE PAPER. THESE HAVE NOW BEEN COMPLETED AND THE EXPERIMENTAL DATA INCORPORATED INTO FIGURE 9. FIGURE 9 HAS ALSO BEEN MODIFIED BY THE ADDITION OF COLOR HIGHLIGHTS TO MAKE IT EASIER TO IDENTIFY THE HOMOLOGOUS SEQUENCES. BECAUSE WE HAD ADDITIONAL DATA FROM PREVIOUS STUDIES REGARDING INSULIN BINDING REGIONS OF THE INSR, WE ALSO TESTED THESE OTHER REGIONS OF THE INSR FOR COX ANTIBODY BINDING – THUS THE ADDITIONAL SEQUENCES THAT WERE TESTED. NOTABLY, REGIONS THAT BIND INSULIN ALSO CORRELATED WELL WITH COX ANTIBODY BINDING, WHICH IS AGAIN COLOR HIGHLIGHTED IN THE NEW FIGURE 9.

Do Figures 11-13 really contribute to what is already shown in figure 9 and 10? Can’t this be moved to supplemental?

GENERALLY, ONE PROVIDES AT LEAST SOME EXPERIMENTAL RESULTS TO VALIDATE OR ILLUSTRATE THE TYPE OF DATA UPON WHICH TABULAR RESULTS ARE PROVIDED. HOWEVER, AS REQUESTED, WE HAVE MOVED MOST OF FIGURES 11-13 TO AN APPENDIX.

As similar approach may be taken for figures 14-18.

DITTO

Figure 19 to 21 could be combined to one figure.

PARTIALLY COMBINED AND PARTIALLY MOVED TO AN APPENDIX

The experiments in section 2.7 only involve a very limited number of patients, moreover each figure appears to have a different set of patients.

WE HAVE ADDED A TABLE THAT SUMMARIZES ALL OF THE EXPERIMENTS RUN ON ALL OF THE PATIENT SERA. WE AGREE THAT THE NUMBER OF PATIENT SERA TESTED IS LIMITED BUT GIVEN THAT THE STUDY HAD NO EXTERNAL FUNDING, WE WERE SEVERELY LIMITED IN THE REAGENTS AND SERA THAT WE COULD AFFORD. THIS IS THE PROBLEM WITH DOING RESEARCH THAT PEER REVIEWERS DON’T CONSIDER FUNDABLE!

Figure 26 does not show T2 diabetics although the legend indicates this.

T2 DIABETICS WERE MENTIONED IN ERROR – NOW CORRECTED.

Many traces do not have error-bars and I don’t see any statistics to support significant findings.

ERROR BARS WERE OMITTED BECAUSE THE MAKE MOST OF THE FIGURES UNREADABLE. HOWEVER, THEIR INCLUSION WOULD MAKE NO DIFFERENCE IN THE INFLECTION POINTS OF THE CURVES SINCE ANY VARIATION IS IN THE Y AXIS VALUE WHILE THE ONLY IMPORTANT ASPECT OF THE CURVES IS IN THE X AXIS (CONCENTRATION) AND ONLY CURVES THAT HAD VERY CLEAR SIGMOIDAL SHAPES WERE USED TO GENERATE BINDING CONSTANTS. WE HAVE TESTED THIS IN PREVIOUS PAPERS.

The idea that Clostridia antibodies will be complementary to COX antibodies and will form idiotype-anti-idiotype pairs that “confuse” the immune system is an interesting hypothesis to explain why perhaps two infections are needed to develop autoimmunity. How would this confusion work? The authors speculate elaborately on this in the discussion and I find this too speculative in the absence of any experimental work proving that a coinfection of clostridium and COX is indeed sufficient to induce T1DM. I think that the discussion can be significantly reduced in size.

WE HAVE ELIMINATED ABOUT HALF OF THE DISCUSSION OUTRIGHT AND TRIMMED THE REMAINING TEXT.

Submission Date

27 March 2023

Date of this review

06 Apr 2023 17:28:42

Reviewer 2 Report

Root-Bernstein and colleagues look to explore the hypothesis of type 1 diabetes autoimmunity being triggered by co-existent infection (most notably coxsackieviruses and Campylobacter) through respective mimicry of INSR and INS and evidence the cross reactivity of the generated antibodies. I think the topic is highly relevant and the study was well conducted. My main concern was the presentation of the manuscript particularly the length and narrative of the discussion. I think this manuscript would benifit from being condensed and with the discussion more focused on the conducted study

My main comments are

11)   Length of manuscript

Both the results (including the number of figures (32!)) and Discussion suffer from being (in the opinion of this reviewer) excessively long and in places repetitive. I think the number of figures should be significantly reduced (? ESM) with more focus on key findings. I think the summary figures were useful (again multiple ? be combined) and easy to follow and could replace much of the more complex description of infection antibody interactions outside of main results (repeated a number of times between summary of results, main results, overview of results at start of discussion and more detailed discussion summary).

22)      Interpretation exceeds data

The discussion is difficult to read both due to the aforementioned length and repetition but also as there is blurring between data and conjecture and significant digression away from the main study. For much of the discussion the reader would be forgiven for thinking they were reading an opinion piece rather than an objective discussion of the experimental data. I also have a few specific comments with regard to some of the statements made which I believe have some factual inconsistencies

  • 1) “Most new T1DM diagnoses are among children”. Over half of t1d has been shown to actually occur in adults (references in PMID: 34670785)

·     2)   “One is that anyone diagnosed with either an  enterovirus infection or a Clostridium infection should be tested for the complementary infection and, if both infections are present, appropriate antibiotic therapy should be im-plemented immediately.” Is there data that antibiotics would significantly reduce the immune response and thus reduce the development of INS/INSR ab? Ref needed if there is.

  • 3) “Additionally, the incidence of new T1DM diagnoses correlates reasonably well with COX incidence.” This suggests that the clinical presentation of type 1 diabetes follows very closely in time with Cox infection as a cause of the AI process. It is well documented that stage 1 of T1D (multiautoantibody development) precedes clinical diagnosis of T1D by many months to years and is the justification for population Ab screening (PMID: 31990315). Thus the fact that peak T1D clinical diagnosis is similar to Cox infection is hard, in this readers mind, to use as any evidence for causation.

Other comments

The article Introduction is written in a more relaxed narrative tone than one encounters in other manuscripts, this is not a criticism as i actually enjoyed this style and found the introduction easy to follow written in this way. That being said I think some more traditional sign posts throughout would be helpful. For instance whilst insinuated I think very clearly stated aims after the introduction would be helpful and improve readability, the jump to results would be better cushioned by this. For the results I read the first paragraph to be a summary, I think this could be more explicitly stated as such rather than “unfold as follows” if kept (see above comments on repitition)

Author Response

Reviewer 2

Top of Form

Open Review

Quality of English Language

( ) English very difficult to understand/incomprehensible
( ) Extensive editing of English language and style required
( ) Moderate English changes required
(x) English language and style are fine/minor spell check required
( ) I am not qualified to assess the quality of English in this paper

Yes

Can be improved

Must be improved

Not applicable

Does the introduction provide sufficient background and include all relevant references?

(x)

( )

( )

( )

Are all the cited references relevant to the research?

(x)

( )

( )

( )

Is the research design appropriate?

(x)

( )

( )

( )

Are the methods adequately described?

(x)

( )

( )

( )

Are the results clearly presented?

( )

(x)

( )

( )

Are the conclusions supported by the results?

( )

( )

(x)

( )

Comments and Suggestions for Authors

Root-Bernstein and colleagues look to explore the hypothesis of type 1 diabetes autoimmunity being triggered by co-existent infection (most notably coxsackieviruses and Campylobacter) through respective mimicry of INSR and INS and evidence the cross reactivity of the generated antibodies. I think the topic is highly relevant and the study was well conducted. My main concern was the presentation of the manuscript particularly the length and narrative of the discussion. I think this manuscript would benifit from being condensed and with the discussion more focused on the conducted study

My main comments are

11)   Length of manuscript

Both the results (including the number of figures (32!)) and Discussion suffer from being (in the opinion of this reviewer) excessively long and in places repetitive. I think the number of figures should be significantly reduced (? ESM) with more focus on key findings. I think the summary figures were useful (again multiple ? be combined) and easy to follow and could replace much of the more complex description of infection antibody interactions outside of main results (repeated a number of times between summary of results, main results, overview of results at start of discussion and more detailed discussion summary).

CONDENSED AS REQUESTED. HALF OF THE DATA HAVE BEEN MOVED TO APPENDICES, TWO OF THE SUMMARY FIGURES COMBINED AND THE OTHERS ELIMINATED (THOUGH ONE IS NOW USED AS A VISUAL ABSTRACT FOR THE PAPER).

22)      Interpretation exceeds data

The discussion is difficult to read both due to the aforementioned length and repetition but also as there is blurring between data and conjecture and significant digression away from the main study. For much of the discussion the reader would be forgiven for thinking they were reading an opinion piece rather than an objective discussion of the experimental data.

REVISED TO ADDRESS CONCERNS. IN PARTICULAR, HALF OF THE DISCUSSION HAS BEEN ELIMINATED AND THE REST REVISED FOR BREVITY. THE CONCLUSION IS ELIMINATED.

I also have a few specific comments with regard to some of the statements made which I believe have some factual inconsistencies

  • 1) “Most new T1DM diagnoses are among children”. Over half of t1d has been shown to actually occur in adults (references in PMID: 34670785)

REFRENCE ADDED AND LANGUAGE MODIFIED TO TAKE THIS POINT INTO ACCOUNT. THIS STILL MEANS THAT, PER YEAR, A CHILD HAS ABOUT FIVE TIMES THE LIKELHOOD OF BEING NEWLY DIAGNOSED THAN AN ADULT OVER 18.

  •    2)   “One is that anyone diagnosed with either an enterovirus infection or a Clostridium infection should be tested for the complementary infection and, if both infections are present, appropriate antibiotic therapy should be implemented immediately.” Is there data that antibiotics would significantly reduce the immune response and thus reduce the development of INS/INSR ab? Ref needed if there is.

ANIMAL EXPERIMENTS SO INDICATE AND ARE ALREADY CITED ELSEWHERE IN THE PAPER WITH REGARD TO THE KRV ANIMAL MODEL OF T1DM (REFERENCE [113]). REMINDER OF THESE EXPERIMENTS IS MADE HERE.

            3) “Additionally, the incidence of new T1DM diagnoses correlates reasonably well with COX incidence.” This suggests that the clinical presentation of type 1 diabetes follows very closely in time with Cox infection as a cause of the AI process. It is well documented that stage 1 of T1D (multiautoantibody development) precedes clinical diagnosis of T1D by many months to years and is the justification for population Ab screening (PMID: 31990315). Thus the fact that peak T1D clinical diagnosis is similar to Cox infection is hard, in this readers mind, to use as any evidence for causation.

THE STAGING OF T1DM HAS ONLY BEEN DOCUMENTED IN CHILDREN, NEVER IN ADULTS, AND ONLY AMONG CHILDREN WITHIN HIGH-RISK FAMILIES, WHICH ARE A MINORITY OF CHILDHOOD CASES. THUS, THE REVIEWER OVER-STATES THE CASE. IN ANY EVENT, A SLOW STAGING SCENARIO CANNOT EXPLAIN THE EQUALLY WELL-ESTABLISHED FACT THAT NEW T1DM DIAGNOSES HAVE SEASONAL PEAKS AND TROUGHS (REFERENCES PROVIDED IN DISCUSSION). THIS ISSUE IS NOW DISCUSSED IN MORE DETAIL IN THE CONCLUSION. I ALSO HAVE TO SAY THAT MY EXPERIENCE OF ADULT CASES IS THAT MOST ARE VERY SUDDEN ONSET CLEARLY ASSOCIATED WITH A PRECEDING INFECTION WEEKS BEFORE DIAGNOSIS – THIS IS, AFTER ALL, HOW COXSACKIEVIRUS INFECTIONS BECAME ASSOCIATED WITH T1DM ONSET IN THE FIRST PLACE!

Other comments

The article Introduction is written in a more relaxed narrative tone than one encounters in other manuscripts, this is not a criticism as i actually enjoyed this style and found the introduction easy to follow written in this way. That being said I think some more traditional sign posts throughout would be helpful. For instance whilst insinuated I think very clearly stated aims after the introduction would be helpful and improve readability, the jump to results would be better cushioned by this. For the results I read the first paragraph to be a summary, I think this could be more explicitly stated as such rather than “unfold as follows” if kept (see above comments on repitition)

SIGN-POSTING ADDED AND TWO NEW PARAGRAPHS ADDED TO INTRODUCTION THAT (HOPEFULLY) BETTER EXPLAIN THE LOGIC OF THE PAPER.

Submission Date

27 March 2023

Date of this review

30 Mar 2023 14:25:54

Bottom of Form

© 1996-2023 MDPI (Basel, Switzerland) unless otherwise stated

Round 2

Reviewer 1 Report

Authors have adequately addressed my points